# A near-continuous archaeological record of Pleistocene human occupation at Leang Bulu Bettue, Sulawesi, Indonesia

Basran Burhan[1,2☺*], Budianto Hakim[2], Iwan Sumantri[2], Suryatman[2,3], Andi Muhammad Saiful[2,3], Adhi Agus Oktaviana[5,6,7], Ratno Sardi[4,8], Hasliana[8], Muhammad Ramli[8], Linda Siagian[9], Andi Jusdi[10], Abdullah[10], Fardi Ali Syahdar[11], Hamrullah[12], Imran Ilyas[10], Putra Hudlinas Muhammad[7,13], Sofyan Setia Budi[8], Nur Ihsan Djindar[2,3,14], Shinatria Adhityatama[5,15], Rustan Lebe[2,16], Marlon N. R. Ririmasse[15], Irfan Mahmud[2,4], Akin Duli[2,3], Yinika L. Perston[5,17], Mark W. Moore[18], Mariana Sontag-González[19,20], Bo Li[19,20], Gerrit D. van den Bergh[21], Maxime Aubert[5,17,22], Rainer Grün[1,23,24,25], David P. McGahan[1], Michelle C. Langley[1,23], Emma C. James[26], Tiina Manne[27], Ian Moffat[28], Brian Jones[29], Adam Brumm[1,23☺*]

1 Australian Research Centre for Human Evolution, Griffith University, Brisbane, Australia, 2 Pusat Kolaborasi Riset Arkeologi Sulawesi (BRIN-Universitas Hasanuddin), Makassar, Indonesia, 3 Departemen Arkeologi, Fakultas Ilmu Budaya, Universitas Hasanuddin, Makassar, Indonesia, 4 Pusat Riset Arkeologi Prasejarah dan Sejarah, Badan Riset dan Inovasi Nasional (BRIN), Indonesia, 5 School of Humanities, Languages and Social Sciences, Griffith University, Queensland, Australia, 6 Pusat Riset Arkeometri, Organisasi Riset Arkeologi, Bahasa dan Sastra, BRIN, Indonesia, 7 The National Museum of Ethnology, Japan, 8 Indonesian Association of Archaeologists (IAAI) Komda Sulawesi, Maluku and Papua, Indonesia, 9 Museum Kepresidenan Republik Indonesia Balai Kirti, Bogor, Indonesia, 10 Balai Pelestarian Kebudayaan Wilayah XIX, Makassar, Indonesia, 11 Yayasan Bumi Toala Indonesia, Makassar, Indonesia, 12 Korps Pencinta Alam Universitas Hasanuddin, Makassar, Indonesia, 13 Muséum National d'histoire Naturelle, France, 14 Antwerp Cultural Heritage Sciences, Faculty of Design Sciences, University of Antwerp, Belgium, 15 Pusat Riset Arkeologi Lingkungan, Maritim, dan Budaya Berkelanjutan, Organisasi Riset Arkeologi, Bahasa, dan Sastra, BRIN, Indonesia, 16 Badan Layanan Umum Museum dan Cagar Budaya, Direktorat Jenderal Kebudayaan, Jakarta, Indonesia, 17 Griffith Centre for Social and Cultural Research, Griffith University, Australia, 18 Archaeology and Palaeoanthropology, University of New England, Armidale, Australia, 19 Centre for Archaeological Science, Environmental Futures, University of Wollongong, Wollongong, Australia, 20 Luminescence Dating Laboratory, Research Laboratory for Archaeology & the History of Art, School of Archaeology, University of Oxford, United Kingdom, 21 Australian Research Council Centre of Excellence for Australian Biodiversity and Heritage, University of Wollongong, Wollongong, Australia, 22 Geoarchaeology and Archaeometry Research Group, Southern Cross University, Lismore, New South Wales, Australia, 23 School of Environment and Science, Griffith University, Brisbane, Australia, 24 Research School of Earth Sciences, The Australian National University, Canberra, Australia, 25 School of Geography, Nanjing Normal University, Nanjing, China, 26 Everick Archaeology and Cultural Heritage Australia, Brisbane, Australia, 27 School of Social Science, Faculty of Humanities, Arts and Social Sciences, University of Queensland, Brisbane, Australia, 28 College of Humanities, Arts and Social Sciences, Flinders University, Adelaide, Australia, 29 School of Earth, Atmospheric and Life Sciences, University of Wollongong, Wollongong, Australia

☺ These authors contributed equally to this work.
* basran.basran@griffith.edu.au (BB); a.brumm@griffith.edu.au (AB)

## Abstract

Prior research has indicated that the Indonesian island of Sulawesi was host to archaic hominins of unknown taxonomic affinity from at least 1.04 million years ago (Ma), while members of our own species (*Homo sapiens*) were probably established on this Wallacean landmass from at least 51.2 thousand years ago (ka), and possibly as early as 65

**Data availability statement:** All relevant data are within the paper and its Supporting Information files.

**Funding:** Australian Research Council grants DE130101560 and FT160100119 awarded to A.B. and DE160100703 awarded to I.M., with additional support provided by the Wenner Gren Foundation (Post-Ph.D. grant) and Griffith University.

**Competing interests:** The authors have declared that no competing interests exist.

ka. Despite this, the paucity of well-dated Pleistocene archaeological sites from Sulawesi means that very little has been known about the pattern and timing of early human occupation of the island, including whether there is any evidence for overlap between archaic hominins and modern humans, and when and how the former went extinct. Here, we report the results of multiple seasons of deep-trench excavations at Leang Bulu Bettue, a limestone cave rock-shelter complex in the Maros-Pangkep karst region of South Sulawesi. Leang Bulu Bettue is the only site presently known on the island with an archaeological record ranging in age from the Middle to Late Pleistocene to late Holocene periods. Investigations at this site since 2013 have revealed an extensive sequence of stratified deposits down to a depth of about 8 m below the surface. Notably, there is evidence for animal butchery and stone artefact production including a stone 'pick' at around 132.3–208.4 ka followed by a major shift in human cultural activity during the Late Pleistocene. By around 40 ka, an earlier occupation phase (Phase I) characterised by a straightforward cobble-based core and flake technology and faunal assemblages dominated by extant dwarf bovids (*Bubalus* sp., anoas), but including now-extinct proboscideans, had been replaced by an entirely new occupation phase (Phase II) with a markedly distinct archaeological signature, including the first evidence for artistic expression and symbolic culture. We consider the implications of this behavioural disconformity for our understanding of the history of humans on Sulawesi, including the possibility it reflects the replacement of archaic hominins by modern humans.

## Introduction

The Indonesian island of Sulawesi is the world's eleventh largest island, and the fourth largest in Indonesia after New Guinea, Borneo and Sumatra. It is the most extensive terrestrial habitat in the Wallacean archipelago ("Wallacea"), a unique biosphere comprising ~8000 oceanic islands scattered between the Southeast Asian continental shelf (Sunda) and the Pleistocene low-sea level landmass of Australia-New Guinea (Sahul) [1–2] (Fig 1). Owing to long-term isolation the Wallacean islands are characterised by unbalanced and impoverished mammalian faunal diversities [5]. Sulawesi was only 50 km to the east of the adjacent Southeast Asian mainland (present-day Borneo) at times of −120 m sea-level, but it is separated from Sunda by the ~2000-m-deep Makassar Strait, a formidable zoogeographic barrier. Host to the most diverse endemic fauna in Wallacea [6], the island's mammal community is predominately Asian in affinity and exhibits a high degree of endemism (~92–98%, excluding bats), demonstrating that Sulawesi was a difficult colonisation prospect for non-flying Asian land mammals [6–10]. Nonetheless, pigs, bovids, squirrels, civets, and primates (tarsiers and macaques) all reached Sulawesi from source populations in Sunda, but were unable to colonise other parts of Wallacea [6,8], suggesting this island was the most frequent recipient of sweepstakes dispersals from adjacent mainland Asia [11]. The living endemics are all primitive and ancient lineages within their clades [9,12].

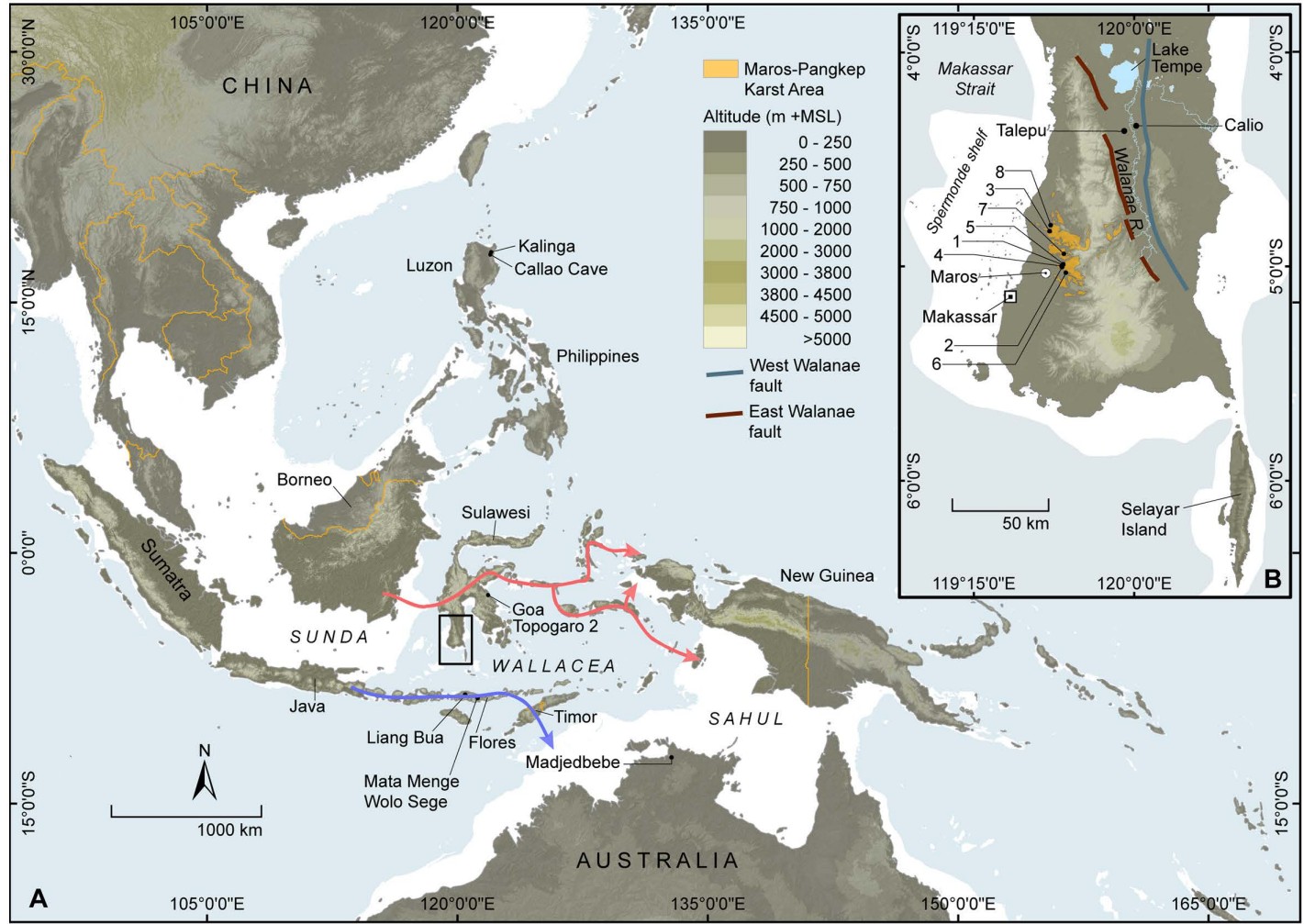

**Fig 1. Map of the Maros-Pangkep study area. A**, Sulawesi in its regional context. It is the largest landmass in Wallacea, the zone of biofaunally unique oceanic islands located between the continental regions of Sunda and Sahul. The two main conjectured dispersal pathways taken by modern humans to get from Sunda to Sahul are represented by red (northern route) and purple (southern route) arrows. The southwestern peninsula of Sulawesi (area within the rectangle) is shown in the inset panel **(B)**. **B**, South Sulawesi, showing the location of the Maros-Pangkep limestone karst area and sites mentioned in the paper: 1) Leang Bulu Bettue; 2) Leang Burung 2; 3) Leang Sakapao 1; 4) Leang Timpuseng; 5) Ulu Leang 1; 6) Leang Karampuang; 7) Leang Tedongnge; 8) Leang Bulu' Sipong 4. The Walanae Depression is situated between two major faults and bounded to the north by Lake Tempe. Plio-Pleistocene fossil deposits occur on either side of the Walanae River. Basemap adapted from GEBCO [3] available for public use, DENMAS 2025 CC BY_NC 4.0 (https://tanahair.indonesia.go.id/demnas/#/demnas), and reference [4].

It has also become apparent over recent years that this island holds considerable significance for our understanding of early human dispersals and ancient hominin diversity in Southeast Asia and the wider region [5]. In particular, the large size of Sulawesi (~174,000 km$^2$) and its central location and proximity to the Asian continental landmass suggest that it may have been the key conduit through which various hominin taxa dispersed from Sunda into Wallacea and beyond [5]. Sulawesi is positioned along one of the two most likely routes used by early modern humans (*Homo sapiens*) as they dispersed eastward from Sunda to Sahul [13–16] (Fig 1). The initial peopling of Sahul is thought to have taken place by at least 50,000 years ago (50 ka) [17,18], and possibly up to 65 ka based on the age of the earliest excavated archaeological deposits at Madjedbebe in northern Australia ( [19] – but see [18,20,21] for critical discussions rejecting this date].

Outside of the South Sulawesi province, only two archaeological sites with published Pleistocene dates are reported on the rest of the island. The oldest, Goa Topogaro 2, yielded a $^{14}$C age of 42 kyr (thousand years) cal BP for the deepest cultural remains recovered thus far [22] (for further information on this site, see also [23–24]). Presently, the oldest proxy evidence for modern humans in Sulawesi is a rock painting dated using the uranium-series (U-series) method to at least 51.2 ka at Leang Karampuang, a limestone cave in the Maros-Pangkep karsts in the island's south [25]. The imagery is interpreted as a composed scene (for critical discussions of this term, see [26–28]) that depicts three human-like figures interacting closely with a wild pig [25]. This artwork appears to comprise a very rare early figurative representation of a narrative; that is, a pictorial record of storytelling. Based on the relative sophistication of this painting, and the lack of evidence for both figurative depiction and scenic representation in art by species other than our own, the dated artwork at Leang Karampuang has been attributed to cognitively modern *H. sapiens* [25]. The early cave art at this locality is not an isolated occurrence: there are nine other sites in Maros-Pangkep with parietal motifs of various kinds that are dated using U-series to between at least 17 ka and 48 ka [25,29–31].

It is also evident that pre-*H. sapiens* hominins extended their geographic range to the east of the Sunda Shelf, somehow making deep-water crossings—probably accidentally (that is, without forward planning and the use of purposefully constructed watercraft [32–33])—to establish a presence on at least two Wallacean islands, Flores and Sulawesi. On Flores the fossil remains of the endemic, small-bodied hominin species *H. floresiensis* from Late Pleistocene deposits (~100–50 ka) have been recovered at Liang Bua cave [34–36]. A small number of much older *H. floresiensis* fossils [37–38] have also been excavated at Mata Menge in the So'a Basin in deposits that date to around 700 ka [39]. Elsewhere in the So'a Basin, stone tools dating to at least 1.02 ± 0.10 million years ago (Ma) have been excavated at Wolo Sege [40]. Based on the anatomy of *H. floresiensis*, it is hypothesised that the Flores hominins evolved from a group of early Asian *H. erectus* that became isolated on this Wallacean island before ~1 Ma, thereafter reducing in body size within this insular environment [34,37,38]. These hominins could have initially crossed to Flores from Sulawesi [5]. If so, hominins may have reached the latter island at least 1 Ma if not even earlier, having got across to Sulawesi directly from the Sahul mainland (Borneo) or potentially from the Philippines to the north [5]. Concerning the latter, fossils of an extinct, small-bodied hominin (*H. luzonensis*) have been found in Late Pleistocene deposits at Callao Cave on the oceanic island of Luzon in the northernmost Philippines [41]. Near to this site, excavations at the open site of Kalinga in the Cagayan Valley have revealed *in situ* stone artefacts and the butchered remains of an extinct rhinoceros in sediments dating to ~709 ka [42].

On Sulawesi, the oldest evidence for a hominin presence identified thus far come from the Walanae Depression in the Soppeng district of central South Sulawesi. At the deep-trench site of Talepu an assemblage of *in situ* stone artefacts (*n* = 318) has been dated to between approximately 194 ka and 118 ka [43]. These implements were associated with the fossil remains of extinct megafauna (>44 kg) *Stegodon* and the suid *Celebochoerus* spp.. More recently, a small number of stone artefacts (*n* = 7) recovered embedded *in situ* from lithified deposits at the nearby site of Calio have been dated using a combination of techniques to 1.04 Ma and possibly up to 1.48 Ma [44]. This places the earliest known hominin occupation of Sulawesi within a similar timeframe to that seen on the eastern Wallacean island of Flores.

No hominin fossils have been recovered from the Walanae Basin sites, so the identity of this late Middle Pleistocene population of the Walanae area is unknown [43,44]. Based on the fossil record of the wider region, the most plausible candidate would be *H. erectus* or perhaps a dwarfed, insularised variant of this hominin [32,43]. Denisovan hominins are another possibility. Genomic research suggests that ancestors of present-day Aboriginal and Melanesian (Papuan) peoples interbred with Denisovans [45] prior to the colonisation of Sahul [46–49]. It is presently unclear where in the region this ancient gene flow event took place. However, one possibility is that the geographical range of the Denisovan lineage extended from Siberia to Sunda, and possibly to Wallacea [50]. In light of this scenario, Sulawesi is considered to be one of the most likely locations, if not the most likely, where genetic intermingling between our species and Denisovans could have occurred within Wallacea [51].

At Talepu, the top of the sequence terminates at around 100 ka, so no insight is available from that site into the later pattern of human occupation on the island [43]. This gap in knowledge includes the key issue of whether the earliest inhabitants persisted long enough to have overlapped with *H. sapiens* on Sulawesi, and, if so, whether they encountered our species directly. In Maros-Pangkep, excavations at Leang Burung 2 rock-shelter [52] and the high-level cave Leang Sakapao 1 [53] yielded stratified deposits with evidence for human occupation dating back to 30 ka, and between 30–22 ka, respectively. Renewed excavations at Leang Burung 2 [54,55] deepened the original trench [52], revealing new archaeological evidence for human habitation in a clay layer around 5 m below surface, which has been dated to ~50 ka [54]. According to this investigation, the stone technology in these deep deposits is broadly similar to that at Talepu [43], and faunal remains are dominated by the largest of Sulawesi's still-extant mammalian fauna [54]. It was not possible to extend these most recent excavations at Leang Burung 2 below a depth of 6.2 m owing to groundwater seepage. A large amount of the Late Pleistocene sediments at this site also appears to have been removed by erosional processes. The result is a temporal gap in the Leang Burung 2 sequence between approximately 50 ka and 35 ka, a critical period during which the Talepu-like core-and-flake stone technology disappears from the archaeological record at this locality and early evidence of rock art appears in the area [25,29–31].

There is thus a crucial gap in our knowledge of the early history of human habitation on Sulawesi and the wider region between Sunda and Sahul. In particular, we lack an understanding of when our species arrived on this large Wallacean island and the timing of the extinction or extirpation of the archaic hominin population that had evidently established a presence on Sulawesi long before the earliest appearance of *H. sapiens*. To address these issues, it is necessary to establish a more comprehensive understanding of the archaeological record of hominin occupation on Sulawesi during the Late Pleistocene period (~120–12 ka).

Presently, there is only one archaeological site on the island from which dated evidence for hominin occupation covering this approximate time span is available: the limestone cave and rock-shelter complex of Leang Bulu Bettue (hereafter, LBB) in Maros-Pangkep (Fig 1). Prior work at this site has described human occupational evidence gleaned from excavations in the uppermost portion (~0–2 m) of the deposits, focusing on an aceramic cultural phase (here denoted Phase II, superseding the cultural phase designations proposed in [56 and followed in 57]) that is estimated to date to between approximately 40 ka and 16 ka [56–60]). The richest cultural deposits within this occupation phase date to around the time of the LGM. These strata have yielded a relatively complex lithic technology [57], a partial *H. sapiens* maxilla [56], and an assemblage of symbolic material culture items that includes a small number of examples of portable art [58,60] as well as personal bodily adornments manufactured from animal bones and teeth [58]. Extensive evidence for the processing and use of mineral colourants was also uncovered in the LGM deposit and an earlier stratigraphic context (~40 ka) [58]. Key findings include used ochre pieces, ochre stains on stone tools, and a bone tube that may have been used for creating hand stencil art [58]. We attribute the finds in these cultural deposits, dating to between about 40 ka and 16 ka, to occupation by the same modern human population responsible for making the Late Pleistocene rock art [58].

Importantly, our ongoing deep-trench excavations at LBB have demonstrated that archaeological evidence for human habitation continues well below the deepest layer associated with this <40 ka occupation phase. Seven annual excavation seasons have now exposed a deeper and older sequence of cultural layers down to a depth of around 8 m. This part of the record at LBB—which we refer to hereafter as the 'deep deposits'—has not previously been described in the literature, although the topmost stratigraphic unit in this sequence (layer 5) and associated cultural findings have been briefly mentioned in published articles [57,58].

In this paper, we present the first full description of the archaeological sequence at LBB excavated thus far, focusing in particular on the previously undescribed deep deposits at this site. We provide an overview of the stratigraphy and site formation processes at LBB and outline evidence for a shift in human cultural behaviour at the site at around 40 ka. We consider the implications of this change in the record of cultural occupation at LBB, both for the pattern of hominin habitation on Sulawesi in particular and the story of early humans in the wider region more generally.

 

## Description of the study site

LBB is a valley-floor entrance cave and rock-shelter complex situated in the foothills of the lowland karst district of Maros-Pangkep (Fig 2; Supporting Information). The karst terrain in this region outcrops from the alluvial plain over an area of around 450 km² between 4°7´S and 5°1´S. It consists of Eocene to middle Miocene marine-deposited limestone (Tonasa Formation) that has been uplifted and eroded into an array of visually spectacular landforms, ranging from plateau-like limestone massifs to freestanding karstic "towers" that occur in clusters or in isolation (inselbergs) [61–62] – indeed, Leang Bulu Bettue roughly translates as 'cave of the tunnel through the hill'. The limestone tower karst terrain extends almost to the western shoreline at its northernmost extent in the Labakkang district. It is bounded to the east by the volcanic mountains of the Western Dividing Range [61].

Located at 4°59'31.18" S by 119°40'5.53" E, and at an elevation of 18 m above sea level, LBB comprises a rock-shelter with a roof height of 15.6 m and an adjoining chamber that is 27.3 m long, 12.6 m wide, and up to 9.2 m high. The rock-shelter is a recess outside and to the south of the cave mouth. It extends in a north-to-south axis along the base of the karst tower for a distance of 30 m. LBB is situated at the base of a limestone hill and is connected to another cave, Leang Samalea, via a 400 m-long cave passage (Fig 3). Active cave formations are present along this passage and there are two large openings (collapse dolines) with diameters of over 10 m that extend to the top of the limestone tower. Rock art is present on the walls of both LBB and Leang Samalea, the former including heavily weathered hand stencils (*n* = 37) of probable Late Pleistocene antiquity [58]. This art is overlaid by black drawings of Austronesian (Neolithic) style, one of which is dated to 1583–1428 cal BP [63]. It has not been possible to date the hand stencils at LBB (or Leang Samalea) due to the absence of associated calcite deposits. We should note that archaeological research had not been undertaken at LBB prior to the commencement of our excavations in 2013.

## Methods

We conducted excavations at LBB annually in 2013–2015 and 2017–2019 and completed another excavation season in 2023 (for further details, see Supporting Information). This work has focused on two trenches that are separated by a 1 m-wide baulk (Fig 3). The first trench is located just inside the entrance to the interior cave chamber and is henceforth referred to as the Cave Mouth Trench. The second trench—the Shelter Trench—is situated 1 m to the south and focuses on the central floor area of the adjoining rock-shelter. The height of the datum above the floor surface of the Cave Mouth Trench varies from 8–16 cm. Combined, the Cave Mouth Trench and the Shelter Trench cover a total surface area of 36 m². Approximately 132.4 m³ of deposit has been excavated from these two trenches over seven field seasons. The archaeological deposits were excavated by stratigraphic layer, using 10 cm-thick arbitrary "spits" to ensure careful control over the excavation and recording process. In cases where a spit cut across more than one stratigraphic unit (layer), the uppermost unit (i.e., stratigraphically youngest) was removed across the entire spit and finds labelled accordingly (i.e., both spit and layer numbers annotated), prior to excavation of the next layer. A closed-sheet timber shoring system was installed in the trench.

### Depositional sequence

Here we describe our understanding of the depositional history of the deep sequence of sedimentary layers exposed over multiple excavation seasons at LBB. Discussion is drawn principally from excavation of the squares located in the Cave Mouth Trench (squares -A1, -A2, A1, A2, B1, B2) and the Shelter Trench (especially squares -C2, -I1, -I1/1, -H1, -H1/1 as the deepest excavated units) (see Supporting Information). Based on our findings thus far, the archaeological record at LBB can be partitioned into at least four distinct phases of human occupation, Phases I-IV, superseding the cultural phase designations proposed previously [56]. We focus attention on a detailed discussion of the strata which we here assign to Phases I and II, especially the former which are described here for the first time in the published literature (Fig 4).

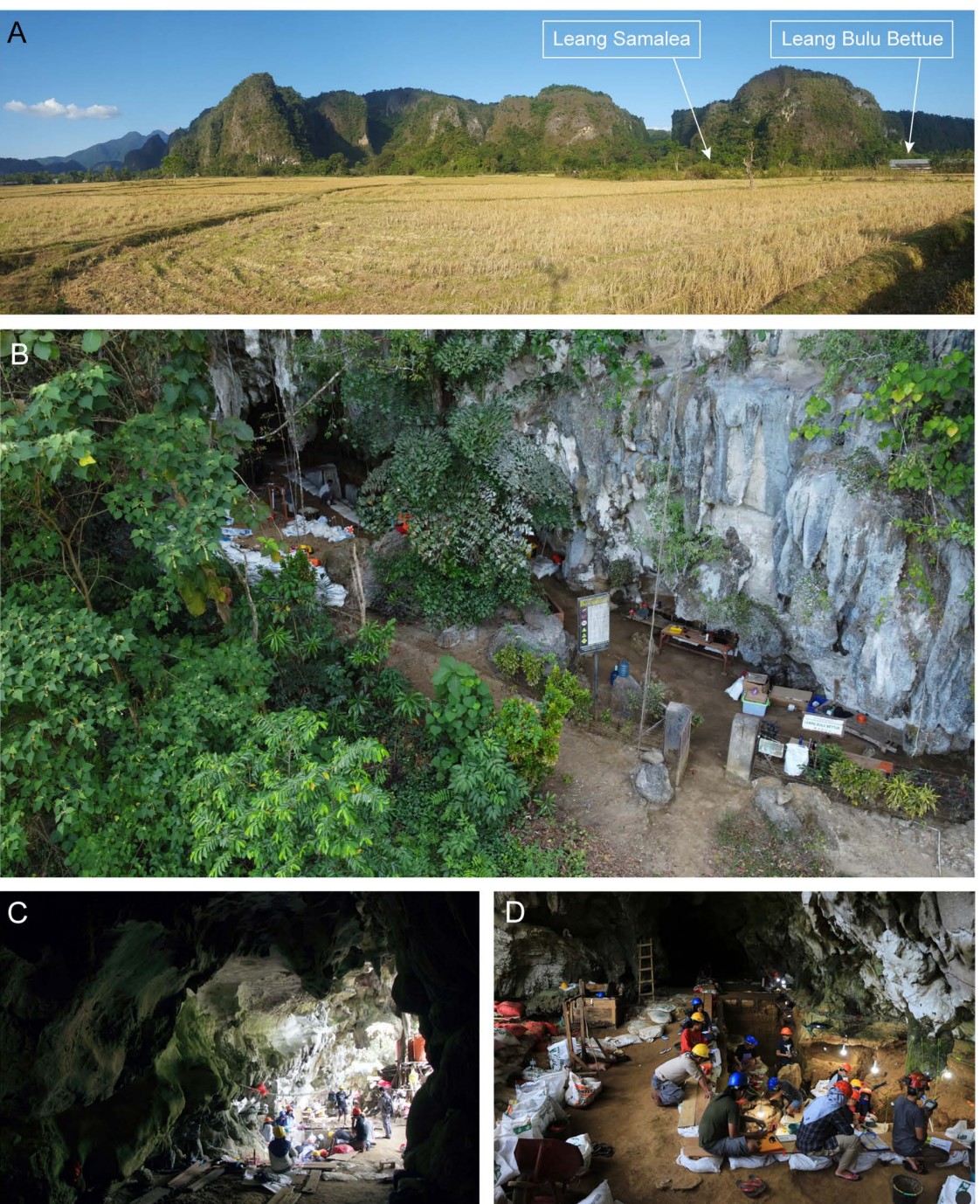

**Fig 2. Leang Bulu Bettue in the Maros-Pangkep karst area of South Sulawesi. A**, Location of the site (and Leang Samalea at the opposite side of the karst tower); **B**, Leang Bulu Bettue cave and rock-shelter complex; **C**, View of cave mouth and adjoining rock-shelter from inside the interior chamber; **D**, Overview of the Shelter Trench (2023), facing north towards the entrance to the interior cave chamber.

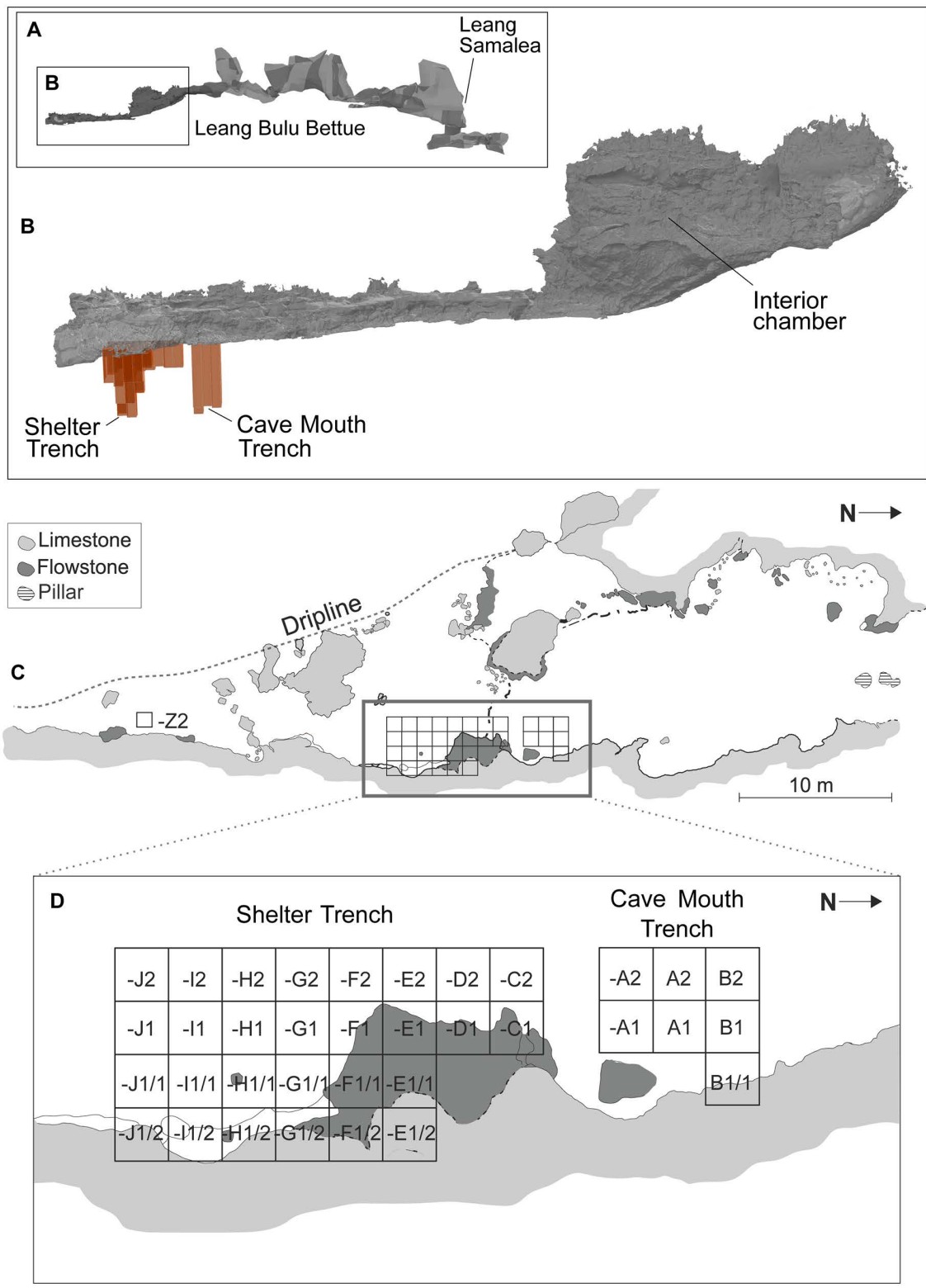

**Fig 3. Leang Bulu Bettue cave. A-B**, 3D rendering of the cave passage through the limestone karst tower showing Leang Bulu Bettue and Leang Samalea at either end (detail of the area inside the red rectangle is shown in **B**). The location of the Cave Mouth Trench and Shelter Trench is shown in **B**; **C-D**, Plan view of Leang Bulu Bettue showing the distribution of the excavation squares (Cave Mouth Trench and Shelter Trench).

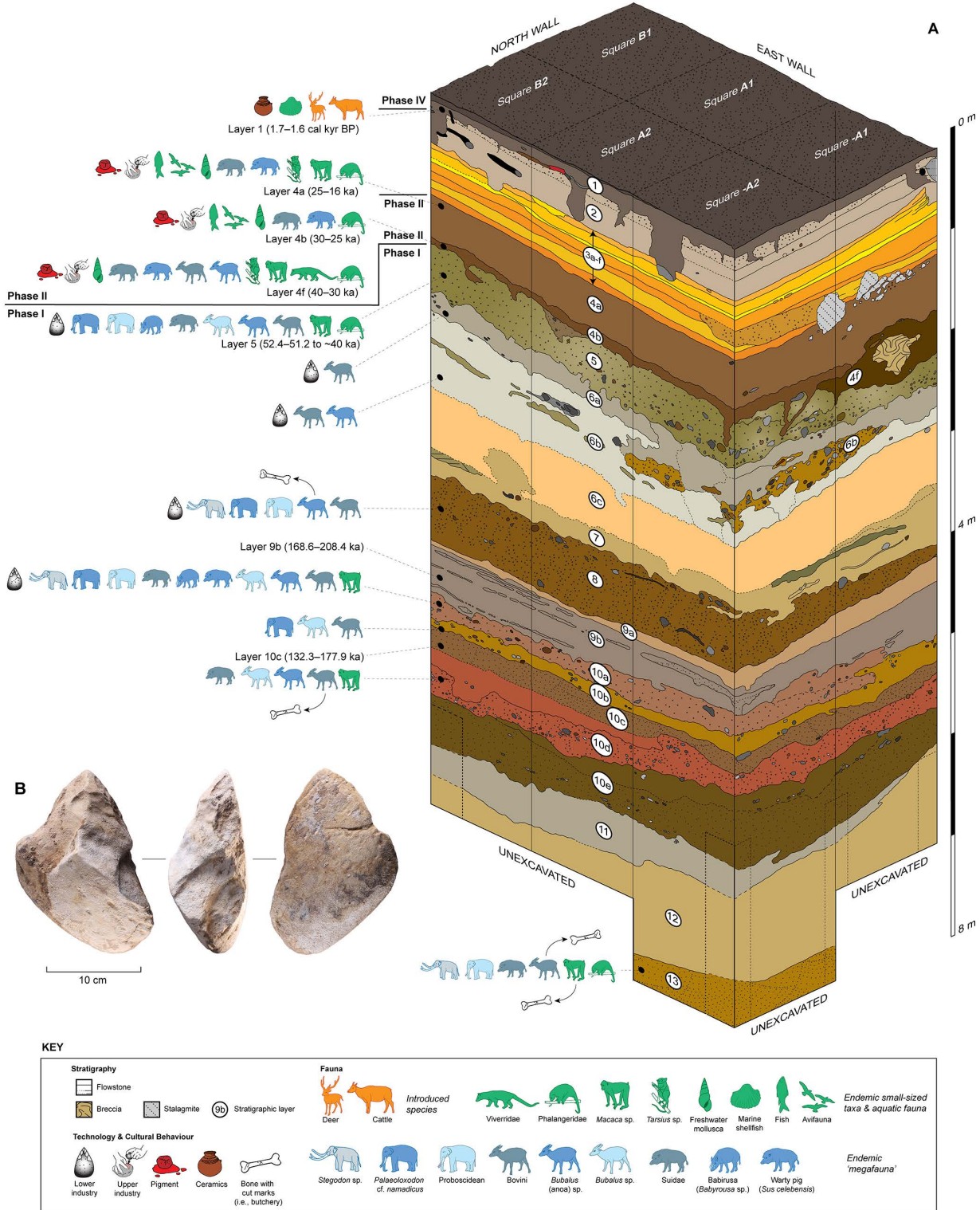

**Fig 4. 3D schematic diagram of the Leang Bulu Bettue chronostratigraphic sequence. A**, Stratigraphy revealed by the Cave Mouth Trench; **B**, deepest stone artefact (unifacial 'pick', layer 10a at 5 m depth) (a 3D model of this artefact is available at: https://une.pedestal3d.com/r/SPiaqbVMjw). Ka = thousand years ago.

## Phase IV

The youngest preserved phase (IV) represents a so-called 'Neolithic' cultural layer (but see, e.g., [64, p.3 & 65] for discussions on the practical problems with applying this term to the island Southeast Asian record) represented by layer 1, a hard-packed moderate brown (5YR4/4) silty clay, and associated rock art of Austronesian style. Dating results indicate that the Phase IV deposits inside the Cave Mouth Trench date to the late Holocene period [58,64]. This unit in square A1 has a mean grain size of coarse to very coarse silt, is poorly to very poorly sorted and has a symmetrical to coarse skewed skewness (see Supporting Information; S3–S4 Fig). The mineralogy is dominated by calcite and sanidine.

A similar sedimentary unit (also denoted layer 1) is also present in the Shelter Trench, but mainly as an intercalation, especially on the eastern side of the excavation squares where it infills a fissure-like cavity or void that formed along the shelter wall. The fissure has a maximum width of 20 cm and follows the shape of the eastern wall, cutting down into the easternmost portions of the underlying layer 4a-e sequence (see below) abutting the wall and possibly also into underlying layer 5. The soil infilling this fissure is generally loose, dry and ashy and it contains recent finds (e.g., pottery fragments) and fresh-looking gastropod shells, clearly distinguishing it from the older layers into which it intrudes. Also present are isolated blocks of cemented yellowish-brown sediment with a silty to sandy texture intercalated with flowstone—redeposited from layers 2 and 3 (see below). Stone artefacts are rare in this disturbed context. This deposit seems to have infilled a recent fissure that was likely caused by water erosion or an undiscovered underground cavity. Other evidence for anthropogenic disturbance includes the modern burial of two cows (one a neonate) and a goat in a hole that was dug from the surface of layer 1 and cut through layer 2. The bottom of this feature reached as deep as the topmost part of the layer 4 sequence.

Layer 1 sits atop a culturally-sterile loose to compact moderate brown (5YR4/5) to dark yellowish brown (10YR4/2) silty clay (layer 2) that is up to 40 cm thick. Postholes originating from layer 1 cut through this layer, terminating at the upper surface of underlying layer 3a. Below layer 2 is a distinctive sequence of cemented flowstones interspersed with coarse pebbly sands to muddy sands that contain high proportions of calcite (corrected weight up to 92.5%) (layers 3a-f). Measuring up to 108 cm in thickness, these well-defined layers vary in colour from moderate yellowish brown (10YR5/4) to dark yellowish brown (10YR4/4). Sloping downwards from the rear of the cave, where they are exposed on the surface as a hard capping flowstone deposit, these layers extend into the rock-shelter, where they generally level out and deepen.

Sediment analysis of layer 3 from squares -C2 and -H2 in the Shelter Trench show mean grain sizes of coarse or medium very poorly sorted sand, although there are some occasional beds with significant fractions of pebble or larger sized grains (representing flowstone units) (see Supporting Information; S1 and S2 Figs). The slightly coarser grain size in square -C2 suggests higher energy depositional conditions may have occurred distal to the rear wall of the cave during this period. Large peaks in $\chi$ and $\chi$fd% observed in both squares are enigmatic, given the culturally sterile nature of the deposit. X-Ray Diffraction (XRD) and Fourier Transform Infra-Red spectroscopy (FTIR) analysis supports the dominance of calcite and clay in these units (see Supporting Information). Layers 2 and 3 in the Cave Mouth Trench are significantly finer grained and very poorly to poorly sorted, with a mean grain size of fine silt to very fine sand. The dominant mineralogy in layers 2 and 3 is calcite and K-feldspar, particularly sanidine. This distinctive sequence of geogenic sediments yielded no archaeological findings or other clear evidence for human occupation. It seems to mark a period in the relatively recent history of LBB when this cave and rock-shelter complex was either uninhabited or there was negligible human activity in this particular part of the site.

## Phase III

Phase III is the Toalean cultural sequence, identified only in a single 1 x 1 m test-pit (square -Z2) excavated in the recess area 15 m to the south of the Shelter Trench. The Toalean is a name given to the regionally distinctive technocomplex marked by the appearance of backed microliths, bone tools, and distinctive stone Maros points around 7 ka [65,66]. The

Toalean deposits at LBB date to between 4582–4829 cal BP [64]. Although found only in a single excavation square, the archaeological finds from this Toalean deposit are distinctive, with the only evidence for backed microliths recovered thus far from LBB [57,64].

### Phase II

Phase II is the part of the Late Pleistocene sequence dating to between roughly 40 ka and 16 ka (Fig 4) (see Supporting Information; S1–S5 Table). It is represented by the layer 4 deposits, a series of strata that is laterally continuous across the Cave Mouth Trench and the Shelter Trench, being represented by layers 4a-b and 4f in the former and layers 4a-e in the latter.

### Phase I

Underlying the Phase II deposits are the Phase I strata, represented by highly distinctive assemblages of cultural and faunal remains recovered from the so-named deep deposits, layers 5–13 in the Cave Mouth Trench and layers 5–12 in the Shelter Trench (Fig 4).

### Stratigraphy and cultural assemblages (Phase II)

Below the sterile geogenic deposits (layers 2–3) is an accumulation of silty clays (layers 4a-b) that increase in thickness at the cave mouth and especially in the central floor area of the rock-shelter, where they inter-mix with localised ashy lenses (layers 4c-d) and reach a combined thickness of 1.5 m at the bottom of the basal layer (4e) (Fig 5) (see Supporting Information). Layers 4a and 4b remain consistent with previously published nomenclature [58]. New stratigraphic observations in squares -G2 and -H2 now suggest that the two previously designated layers 4c and 4d are probably the same stratigraphic unit, with the latter likely constituting a lens within the former. We refer to this as layer 4c/d. Layer 4f has only been encountered inside the Cave Mouth Trench near the eastern end [58].

The layer 4 sequence is thickest in the central floor area of the shelter to the immediate south of an extensive roof-fall pile (see Supporting Information), within the primary light zone (Fig 6). These deposits tilt markedly downward towards the east and tend to thin out to the north of the block-pile and towards the east wall. The rock mass is located on the upper surface of the underlying stratum (layer 5, see below), and it played a key role as a sediment trap and in partitioning the use of space within the site during the Phase II occupation. Excavations in the Shelter Trench show that the layer 4 sequence accumulated on the northern and southern sides of the concentrated mass of large limestone blocks and fallen stalactites. Cavities formed between the rocks also yielded finds within pockets of infilling sediment attributed to this sequence. This extensive roof-fall pile was buried by layers 4a-b.

Layer 4a is a moderate yellowish brown (10YR5/4) slightly sandy "mud" (silt = 50.3%, clay = 32.2%) measuring up to 70 cm in thickness. It can be traced laterally across the full length of the 10 m-long composite stratigraphic section extending from the cave mouth to the central floor area of shelter (Fig 5). Inside the cave, several *in situ* vertical stalagmites had formed on the uppermost surface of layer 4a. We also exposed several large stalactites or stalagmites that had collapsed onto the upper part of this deposit. These observations, and the presence of abundant stone artefacts in the upper few centimetres of layer 4a, followed by a sharp decline in artefact frequencies immediately below, are suggestive of a lag deposit and thus of an erosional event (see below).

Layers 4a-e in the Shelter Trench have yielded the richest cultural remains recovered from LBB (Fig 7). The dense faunal assemblages in layers 4a-e are characterised by shells of *Tylomelania* (formerly *Brotia*) *perfecta* [67], and fragmented and often burnt elements of suids, bear cuscuses (*Ailurops ursinus*), rodents, macaques, and other small mammals (Fig 7). Remains of birds, reptiles, crabs, and fish are also present. The largest animal represented in significant proportions is the Sulawesi warty pig (*Sus celebensis*), a small (40–85 kg), short-legged wild boar that is endemic to the island [68] (Fig 7).

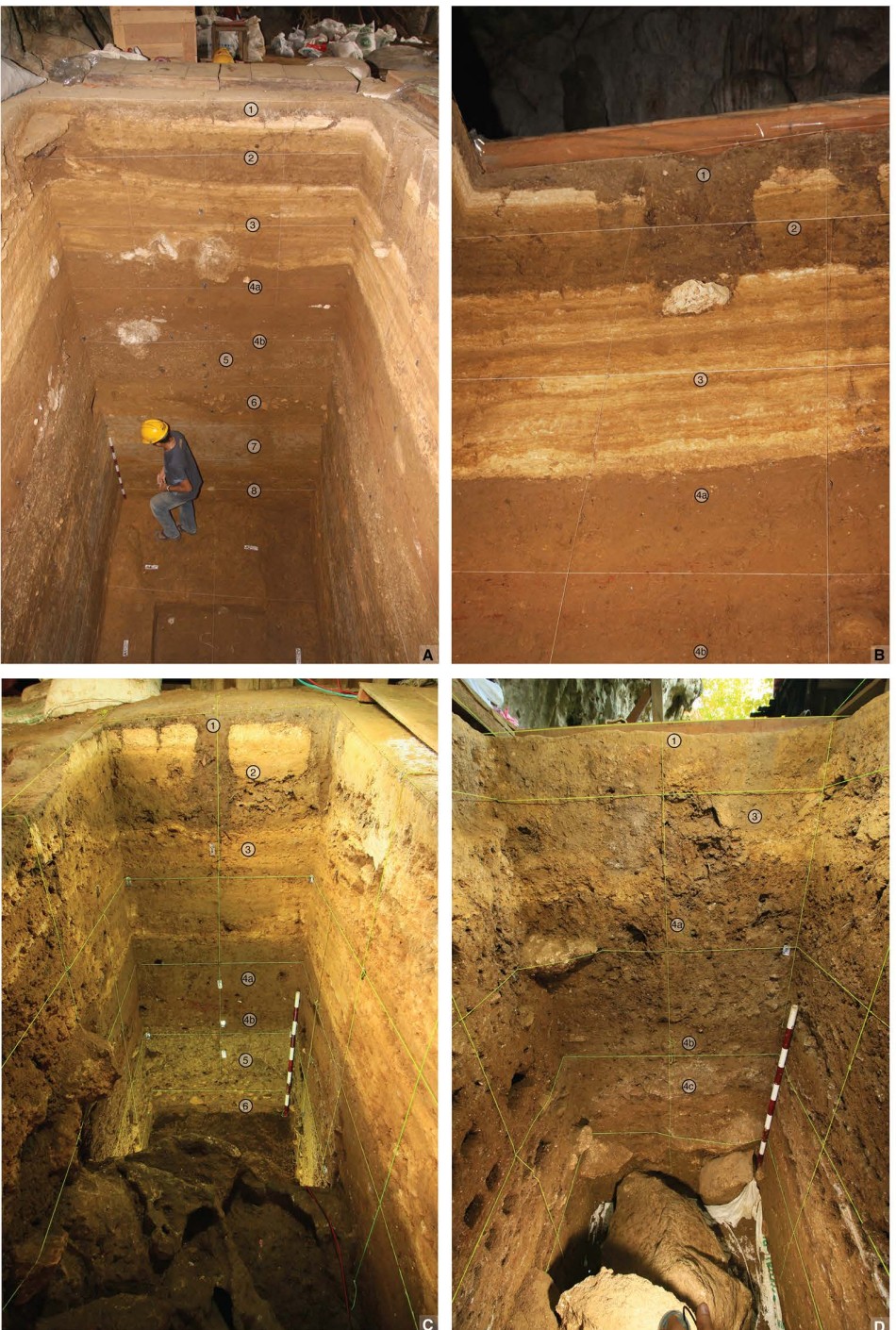

**Fig 5. Stratigraphy at Leang Bulu Bettue. A**, upper stratigraphy to a depth of ~4 m below surface (2014 excavations). The main profile shown is the south wall of squares -A1 and -A2; **B**, west wall of square -A2 (2014 excavations). The distinctive thinly stratified yellowish-orange strata are layers 3a-f (cemented flowstone intercalated with silt). This ~1 m-thick unit lies directly atop the first Late Pleistocene deposit, layer 4a; **C**, north wall of square -C2 (2015 excavations); **D**, south wall of square -H2 (during 2015 excavations).

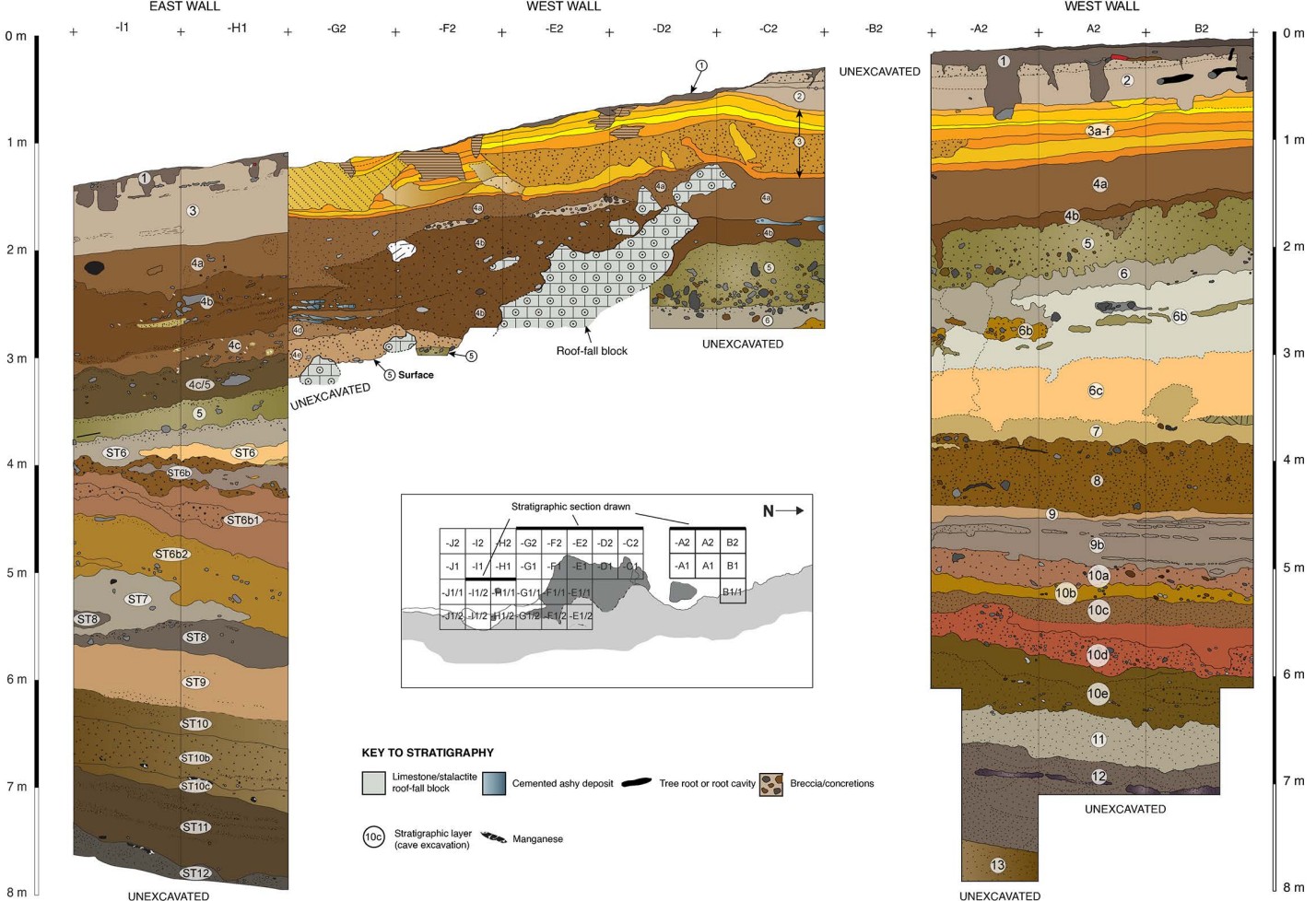

**Fig 6. Stratigraphy at Leang Bulu Bettue.** Depicted here is a composite stratigraphic section exposed in the west walls of the Cave Mouth Trench in the north and the Shelter Trench in the south.

Described in detail elsewhere [57], the Phase II lithic technology involved the common exploitation of chert as a raw material, with relatively minor use of local volcanic and limestone cobbles (Fig 7). Most reduction was conducted using two stone-flaking techniques—direct freehand hard-hammer percussion (Fig 7A, 7F); and well-controlled anvil-supported bipolar percussion (Fig 7B–7D), where the blank was supported on a hard surface and the top edge struck, often initiating flakes from the struck edge as well as the anvil support [57]. Bipolar reduction was a technique for producing sharp-edge flake for use within the cave. Preliminary residue and use-wear analysis of bipolar pieces has indicated that some were also used to process ochre and silica-rich plants [57 p. 12], as were other free-hand stone artefacts at the site. The bipolar cores were small on average, and many were reduced on two axes oriented at 90° to each other (Fig 7B–7D). Freehand, and occasional examples of the bipolar, percussion was often applied to larger, though still small-sized, flake blanks that were themselves produced by the direct freehand hard-hammer percussion technique [57]. Cores for these relatively larger flake blanks were not recovered during our excavations at LBB, indicating that the initial steps in stone tool production were conducted elsewhere in the landscape (e.g., at the quarried source of lithic raw material) or perhaps in another part of the site, although the first scenario seems more parsimonious. Blade-like flakes were also recovered (Fig 7G; Fig 8), including

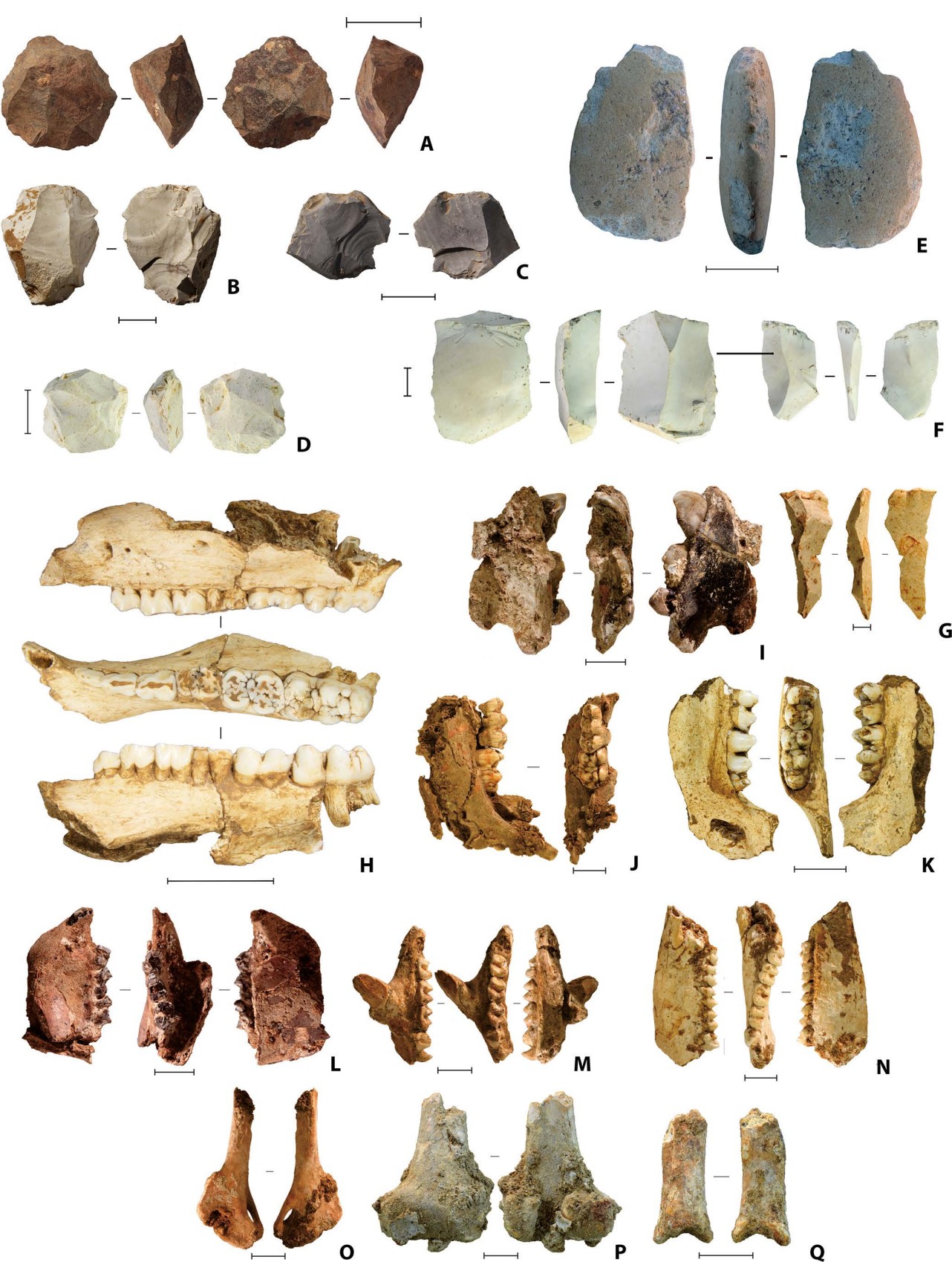

**Fig 7. Typical Phase II faunal remains and lithic artefacts from Leang Bulu Bettue. A**, radial core, square -G2, layer L4c/5 (scale bar 50 mm); **B-D**, chert bipolar artefacts (scale bars 10 mm); **B**, -I1, layer 4a, **C**, -H1 layer 4a, **D**, layer 4a; **E**, anvil made on volcanic rock (scale bar 50 mm); **F**, conjoining flakes from A1 and A2, layer 4, semi-desilicated (scale bar 10 mm); **G**, chert redirecting flake with a second embedded bulb of percussion, layer 4b (scale bar 10 mm); **H**, maxillary and mandibular elements attributed to *Sus celebensis* (Sulawesi warty pig); **I**, layer 4a (scale bar 30 mm); **J** and **K**, layer 4a (scale bars 10 mm); **L**, layer 4a (scale bar 20 mm); **L-N**, maxillary and mandibular elements, *Ailurops ursinus* (bear cuscus) (scale bars 10 mm); **L**, layer 4b, **M** and **N**, layer 4a; **O-Q**, *A. ursinus* postcranial elements; **O** and **P**, layer 4b (scale bar 10 mm), **Q**, layer 4a (scale bar 10 mm).

large macroblades (Fig 8E–8I) similar to those from Leang Burung 2 [52]. However, there is limited evidence at LBB for a formal blade technology, with no definitively prepared blade cores identified [57].

The Phase II occupation yielded the only clear evidence for art and symbolism identified in the Late Pleistocene deposits at LBB [58] (Figs 9–10). This evidence includes utilized ochre pieces, flaked ochre [57], and ochre-stained tools (Fig 8A; 9), personal ornaments in the form of disc-shaped bead blanks made on a suid tooth (Fig 10G), and a drilled bone pendant manufactured from a bear cuscus phalanx (Fig 10H), as well as portable art in the form of stone artefacts incised with abstract markings [58,59,] (Fig 10A–10E), a painted slab (Fig 10F), and a rayed circular engraving [60] (Fig 10I). The earliest evidence for pigment use comprises two ochre-stained chert tools from layer 4f in the Cave Mouth Trench [58] (Fig 9F–9G).

A feature of stone artefacts in Indonesia is the occasional accumulation of silica on tool edges, creating bright gloss patches. This gloss occurs when stone tools are used to process silica-rich plant stems, perhaps for creating thin strips of plant material used in the manufacture of woven material culture items (baskets, mats, and so on) or harvesting foodstuffs [e.g., 69]. Edge-glossed tools are described by Glover [52,70] from Late Pleistocene layers (<30 ka) at Leang Burung 2, and they also occur on Holocene stone artefacts from Flores [71], Timor [72], and Sulawesi [65,73,74]. In total, 133 edge-glossed tools were recovered from a roughly 54% sample of the Phase II deposits at LBB [57].

## Chronology of Phase II

No macroscopically visible charcoal or other carbonized plant material was recovered from the Phase II deposits (see Supporting Information). Dating of these strata was therefore based on a combination of $^{14}$C-dating of plant carbon and freshwater gastropod shells, luminescence-based optical dating methods (feldspar infrared stimulated luminescence, or IRSL), and U-series analysis of *in situ* stalagmites and fossils. The results of the dating program are presented in detail elsewhere [56,58,75]. Briefly, the upper part of the Phase II sequence is well-constrained to between 26 ka and 16 ka by a series of U-series dates obtained from small, in-place stalagmites that formed at various stages during the accumulation of these strata [56,58]. For the basal part of the sequence, laser ablation U-series analyses of bovid teeth from layer 4f ($n = 1$) and the base of layer 5 ($n = 1$) yielded in-sequence minimum ages of 39.8 ± 0.2 ka and 51.8 ± 0.6 ka, respectively [58]. Based on these dates, and other observations which we will discuss below, we infer that initial accumulation of Phase II deposits commenced around 40–39.6 ka [58].

## Phase I stratigraphy

The oldest strata exposed at LBB (from ~2–8 m deep, measured from the modern surface) comprise an extensive series of fluviatile deposits that accumulated to a depth of at least 6 m inside the entrance to the cave (Fig 11). In the Cave Mouth Trench, these deposits are characterised by alternating phases of slack water conditions, leading to the accumulation of archaeologically sterile laminated clays (layers 6c, 7, 9 and 11–12), and the formation of low-energy stream channels that deposited a well-defined sequence of silts, sandy clays, coarse sands, and fine gravels (layers 8, 10a-e, and 13). Capping this sequence of deposits is a 50 cm-thick layer of sandy clay (layer 5), and, below this, lenses of fluvial cobbles in moderate yellowish brown (10YR5/4) slightly sandy clays (layer 6a/b). Layers 8–10 in square A1 have a mean grain size of extremely poorly sorted to poorly sorted medium to coarse grained silt and the dominant mineralogy is sanidine

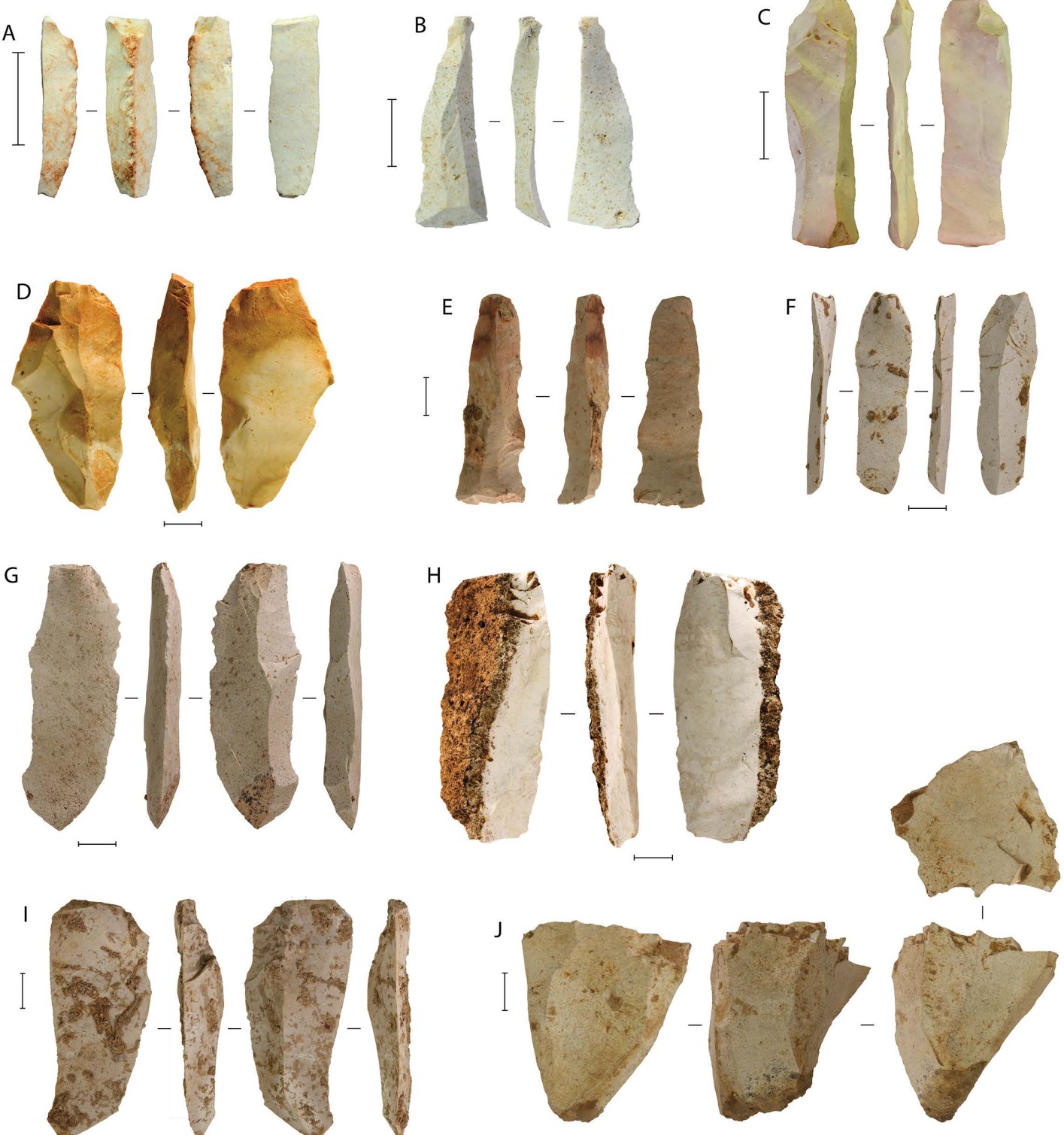

**Fig 8. Chert blade-like flakes from the Phase II occupation. A**, broken blade-like flake with ochre residue on dorsal ridge; **B** and **D**, layer 4b; **C**, **E**, **G**, and **H**, layer 4b; **F**, layer 4/5; **I**, layer 4c; **J**, single-platform blade-like core, layer 4a. All scale bars 10 mm. See also [57] for more information on the lithic technologies at LBB.

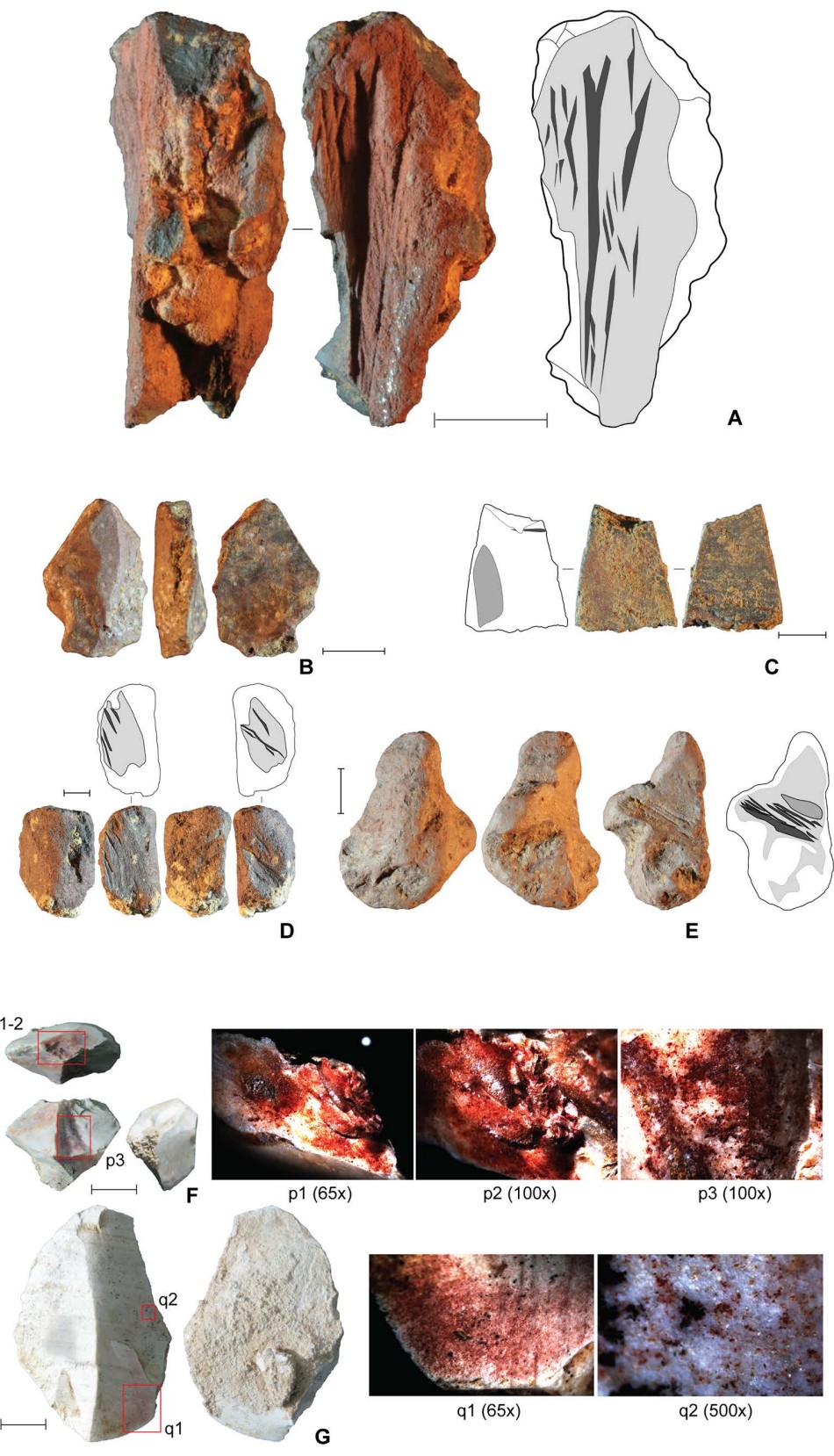

A

B

C

D

E

p1-2

p3

F

p1 (65x)

p2 (100x)

p3 (100x)

q2

q1

G

q1 (65x)

q2 (500x)

**Fig 9. Finds related to the processing and use of mineral pigment (Phase II). A**, utilised ochre nodule (layer 4a); on the accompanying illustration, striations from scraping are depicted in dark grey; **B**, ochre flake with flake scars at the proximal extremity suggesting it was detached from a larger nodule (layer 4b). A central (dorsal) facet reaches 14.8 mm from the distal edge, and, along with a single facet located on both the left and right sides of this central one, displays evidence for abrasion (note: white residue visible in these images is a fungus that developed post-excavation); **C**, ochre plaquette (layer 4b). A single incised line 7.14 mm long was observed on one side. Also observed on the same side of the incision is a lightly scraped section done with a stone tool-edge; **D**, use-worn ochre piece (layer 4a); light grey = ground area; dark grey = scraped area; **E**, use-worn ochre piece (layer 4a); light grey = ground area; dark grey = scraped area; mid grey = scraped area partial worn away by abrasion; **F**, chert artefact with ochre residues (layer 4f); **G**, chert artefact with ochre residues (layer 4f). All scale bars 10 mm).

and other clay minerals (mainly smectite and interlayered smectite), while layer 11 is poorly sorted, has a mean grain size of coarse silt, and the mineralogy is dominated by sanidine (see Supporting Information; S3 Fig).

As has previously been indicated (Supporting Information), it has not been possible to directly correlate some of the deeper strata revealed inside the cave mouth with those in the adjacent shelter owing to the presence of a concentrated area of large roof-fall blocks in the Shelter Trench. The topmost stratum in this sequence, layer 5, is an important marker horizon that can be traced laterally across all excavated squares in the Cave Mouth Trench and most squares in the Shelter Trench (Fig 6), with the exception of those east of -G1/2, -H1/2, I1/2, and -J1/2 due to a subsequent episode of erosion. It should be noted that below layer 5 in the Shelter Trench we use the annotation ST before the layer number (e.g., layer ST7) to denote that the numbered sequence of stratigraphic layers does not necessarily correspond to that inside the Cave Mouth (In some cases it very likely does, however).

Layer 5 exhibits visually distinctive features compared with overlying layers. It is a mottled, brownish to dusky yellow (5Y6/4) deposit with a moist and clayey texture (Figs 6, 10). Thickness ranges from around 50 cm inside the cave mouth to 20–30 cm in the shelter. It has a sloping orientation similar to that of layers 1 to 4a-e (Shelter Trench). Limestone rubble and calcrete nodules (concretions) with a yellowish colour are frequent occurrences in layer 5, as are lenses of blackish manganese sand. This layer also contains lenses of fluvially-deposited cobbles. The mean grain size of layer 5 in the Shelter Trench is poorly sorted medium-grained sand while in the Cave Mouth Trench it is extremely poorly sorted fine-grained silt. The mineralogy of this layer in dominated by sanidine (square A1) and calcite (square -C1). Additionally, layer 5 in the shelter area is distinguished by the extensive pile of large roof-fall blocks that collapsed onto its upper surface (most visible in -D to -J square rows) and were buried by the overlying Late Pleistocene strata (layers 4a-b). The mottling in layer 5 is suggestive of a weathered soil horizon (paleosol).

Stratigraphic observations suggest that after deposition of layer 5 an episode of erosion took place in the central floor area of the shelter (see Supporting Information), with the consequent erosional cut forming a concave depression in the upper portion of this stratum. The result was the formation of a laterally continuous, ledge-like pedestal of layer 5 that banked up against the wall of the shelter (as visible in squares B1/1, and -F1/2 to -J1/2) (Fig 12). This erosional cut in layer 5 was infilled by layer 4e, with these strata overlapping (i.e., interfingering) in places. During this process of erosion some archaeological finds from layer 5 (e.g., faunal elements) were reworked from this deposit and incorporated into layer 4e. In the east of the Shelter Trench, layer 5 becomes progressively thinner as a result of erosion until it disappears completely, such that layer 4e is in direct contact with underlying layer 6/ST6.

As with inside the nearby cave mouth area, below layer 5 in the shelter is a series of fluvial deposits (layers ST6–12) alternating between silty sand, sandy clay, and clay (Figs 13–15). The sequence is at least 4.5 m thick in this area. Layers ST6–7 (further divisible into sublayers ST6b, ST6b1, ST6b2, and ST6b3/7) are predominantly sandy and range from very dark brown (10 YR 2/2) to dark yellowish brown (10 YR 3/6) in colour. The black layers are suspected to result from manganese oxidation. The mean grain size of layer ST6 in square -H2 is very poorly sorted fine sand while the mean grain size in A1 is extremely poorly sorted fine silt and the dominant mineralogy is sanidine. Occasionally, rounded volcanic rock gravel occurs. These fluvial clasts are typically weathered and partially enveloped by thin coatings of calcium carbonate. Layers ST6b, ST6b1, ST6b2, and ST6b3/7 alternate between layers of clay mixed with sand and sandy layers. In certain

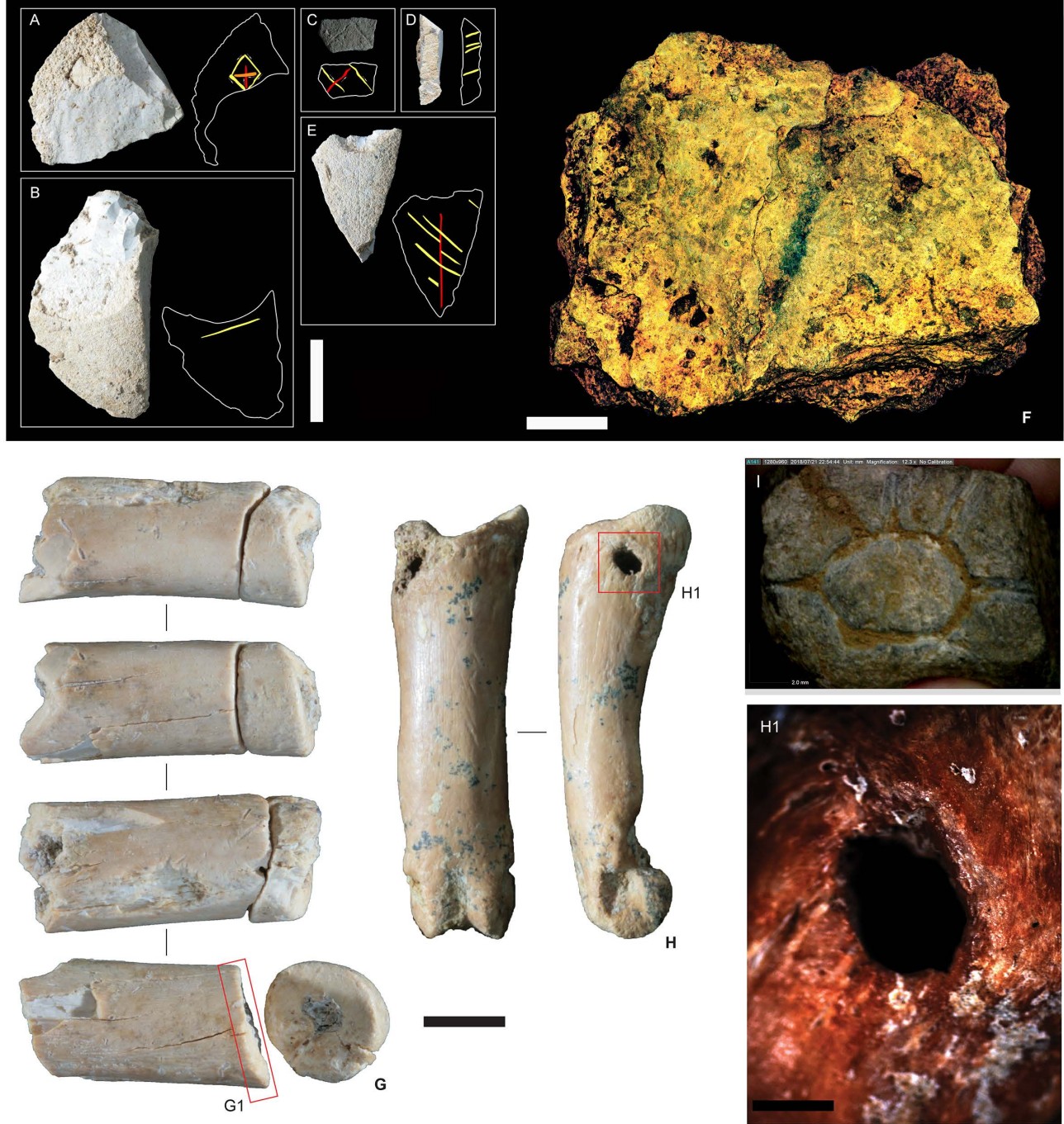

**Fig 10. Symbolic material culture items from Leang Bulu Bettue.** The artefacts were all recovered from Phase II occupation deposits. **A-E**, Stone artefacts with incised lines on remnant cortex patches (vertical white scale bar is 10 mm); **F**, flowstone slab with painted black line; the photograph of the specimen has been enhanced using a *DStretch* filter (YDT) to better show the pigment (horizontal white scale bar is 50 mm); **G**, suid tooth root with refitting bead blank(scale bar 5 mm). **H**, bear cuscus (*Ailurops ursinus*) phalanx with perforation for use as ornament. H1 provides detail of the intact perforation (scale bar 1 mm). **I**, engraved piece of limestone from layer 4a.

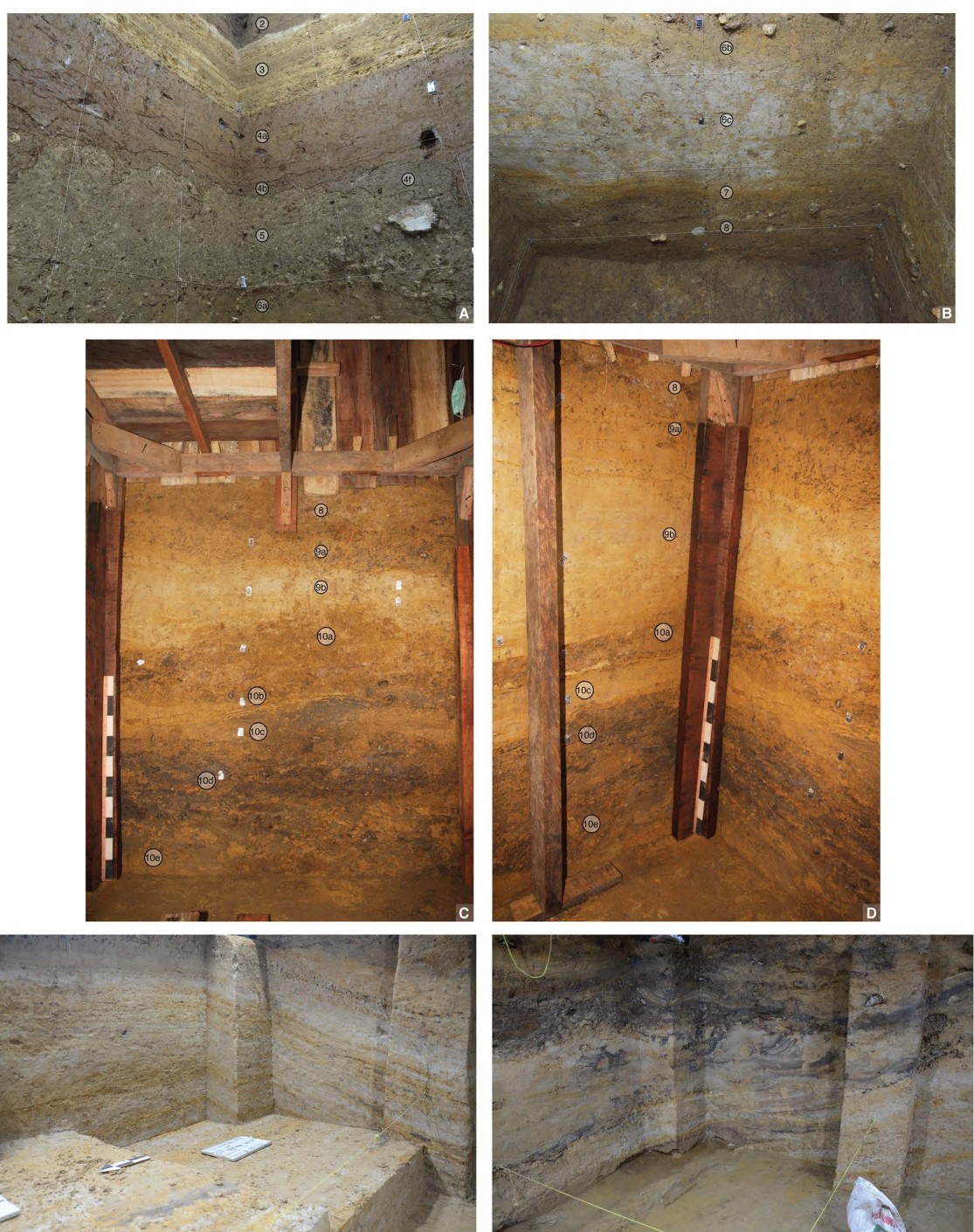

**Fig 11. Lower stratigraphy at Leang Bulu Bettue (Cave Mouth Trench). A**, northern and western walls of the 2014 trench (square B2), showing layers 2 to 6; **B**, southern wall of the 2014 trench (squares -A1 and -A2), showing layers 6a-c and 7-8; **C-D**, layers 8 to 10a-e (2015 excavations); **E**, layers 11-12 (2015 excavations).

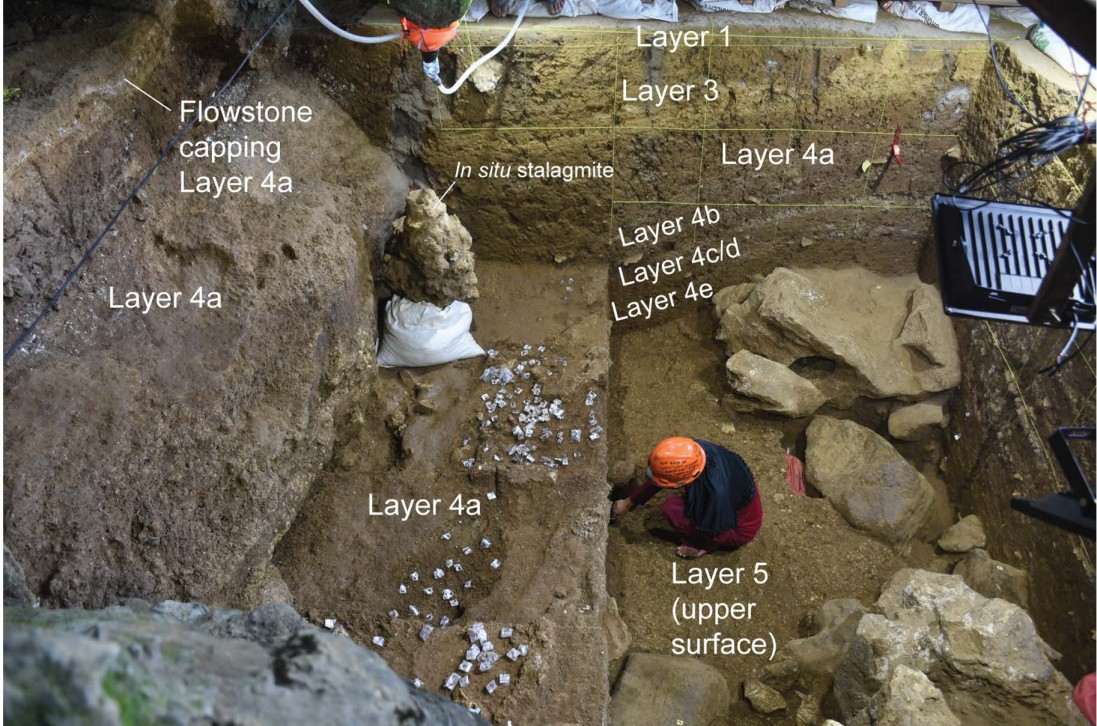

**Fig 12. Stratigraphic sequence exposed in the Shelter Trench (South Wall).** Looking from north to south.

areas, black layers due to manganese oxidation are present, along with concretions with a generally tabular form but containing deposits of loamy sediment in the centre. The latter objects, especially common in layers ST6b and ST6b2, also seem to relate to natural processes of manganese oxidation. Layer ST8 exhibits characteristics like layer ST6b2, consisting of sandy layers interspersed with clay layers. These layers have mean grain sizes in the Cave Mouth Trench of very poorly to poorly sorted very fine to medium silt with a dominant mineralogy of sanidine and other clays. Compared with layer ST6b2, layer ST8 contains more black-coloured soil inclusions. Manganese pisoliths are also more concentrated. Additionally, many pieces of limestone rubble are found. Beneath layer 8 is a yellowish-brown to grey clay horizon (layer ST9) with variable thickness (20–50 cm). It appears to be sterile. Sandy soil lenses are present, becoming more extensive towards the bottom, transitioning into sandy deposits in layer ST10.

Layers ST10, ST10b, and ST10c are predominantly sandy. Layer ST10 still contains many clay inclusions, decreasing in layer ST10b. Layer ST10b consists mainly of dark-coloured sandy layers with almost uniform thickness ranging from 40 to 50 cm. The layer's surface slopes eastward, similar to layers ST8 and ST9 above it. Layer colours vary, with clay tending towards yellowish-brown and sandy soil ranging from reddish-brown to black. Layer ST10b is mostly composed of dark-coloured sand, with some parts being intensely black. Concentrations of manganese are frequently found, sometimes forming thin layers.

Layer ST11 is a clay layer with yellowish-brown and grey colours. It is relatively thick, reaching 50 cm, and becomes thicker towards the northern part of the square -H1. Sandy inserts are found within this layer, ranging from thin (2 cm) to thicker (up to 10 cm) in the middle, thinning out towards the north. The lowest excavated deposit is layer ST12, consisting of a predominantly black sandy matrix. Almost all of layer ST12 is sandy, with no clay inclusions as seen in previous layers, where sand and clay layers overlap. Large amounts of limestone gravel associated with fossilized animal bones or

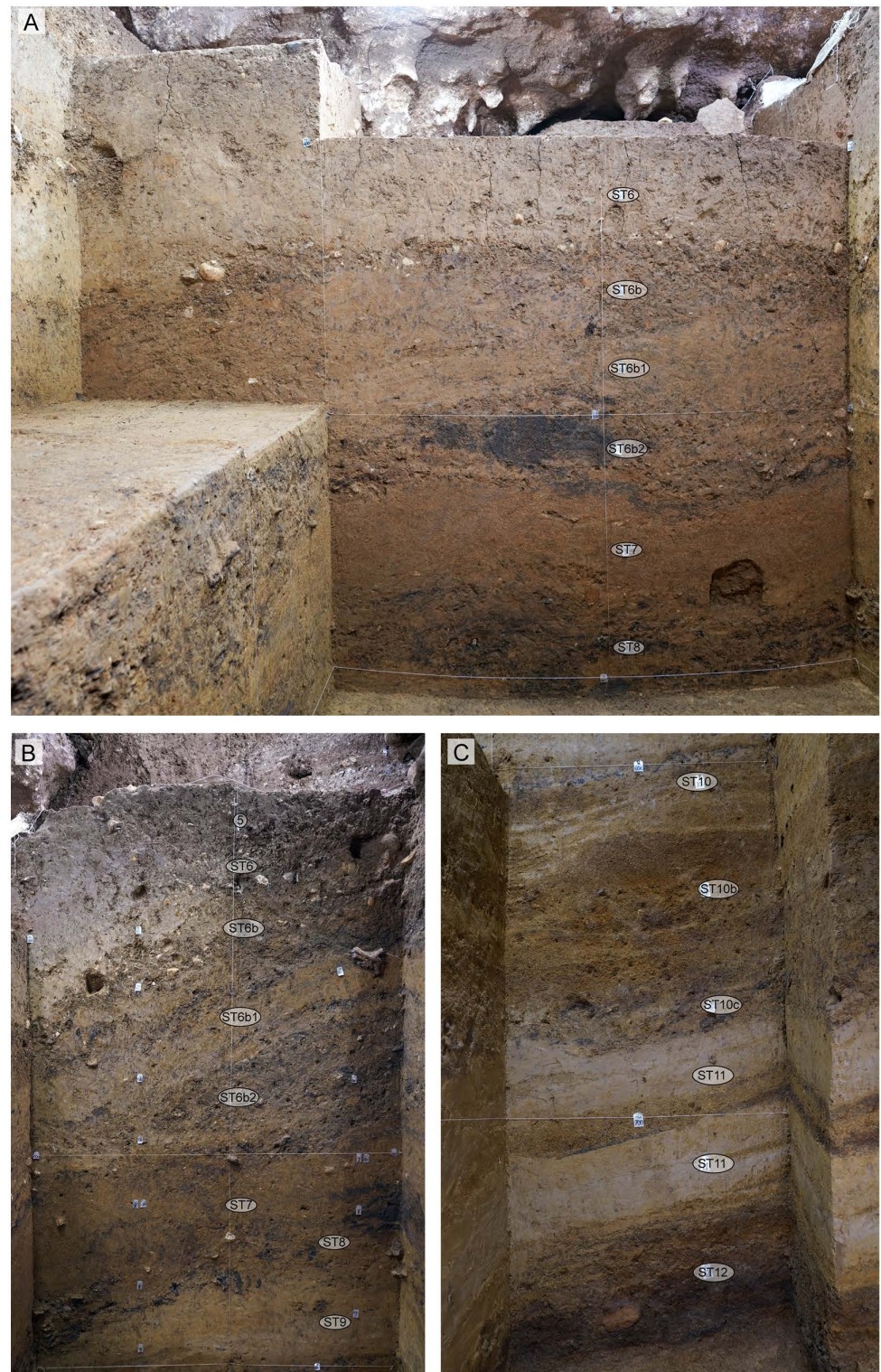

**Fig 13. Lower stratigraphy in Shelter Trench. A**, Eastern wall of squares -G1/1, -H1/1, and -I1/1; **B**, southern wall of the lower part of square -I1; **C**, upper part of the southern wall of squares -I1 and -I1/1.

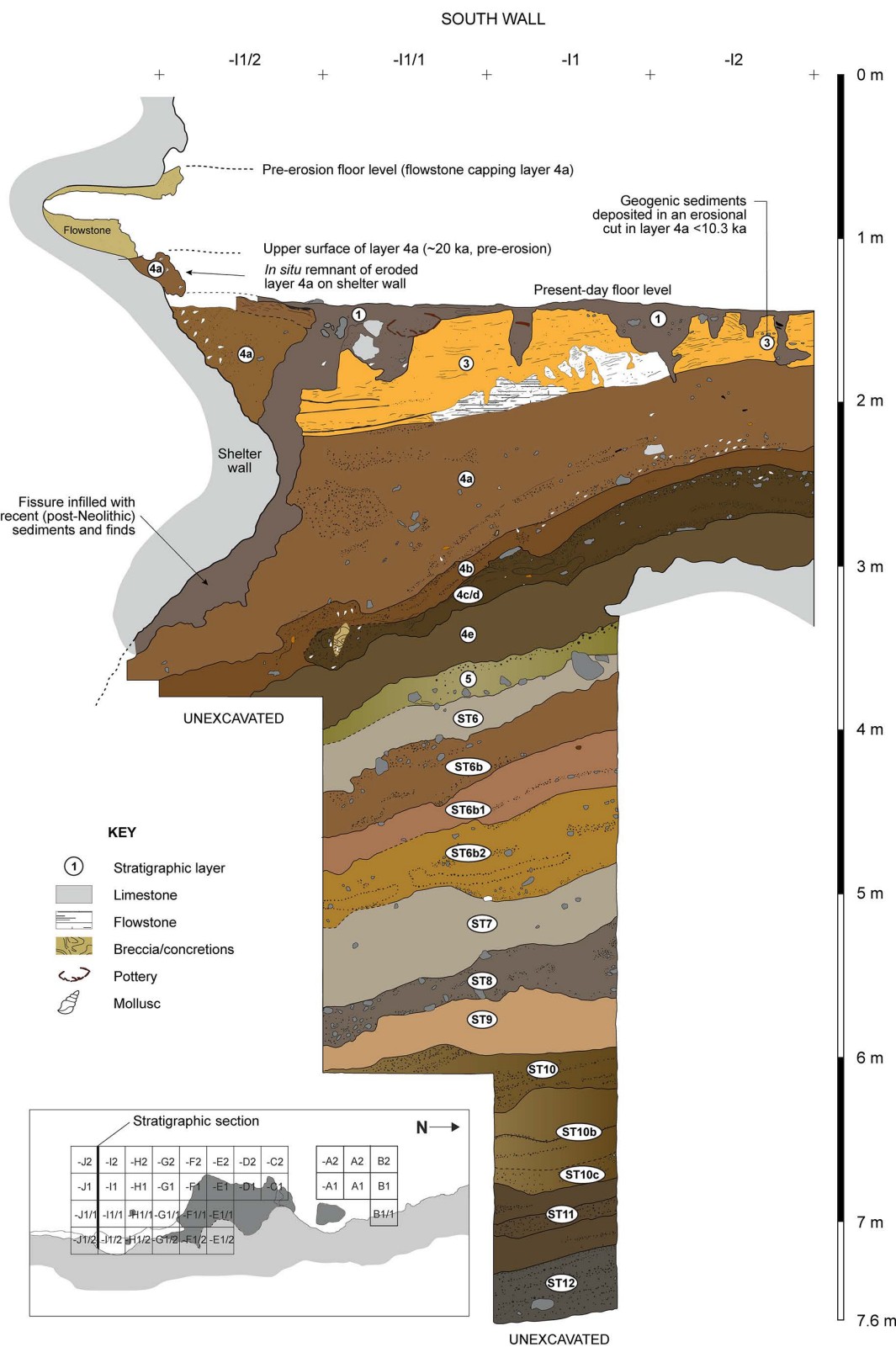

**Fig 14. Stratigraphic sequence at Leang Bulu Bettue.** This illustration shows the lower stratigraphy exposed by excavations in the Shelter Trench.

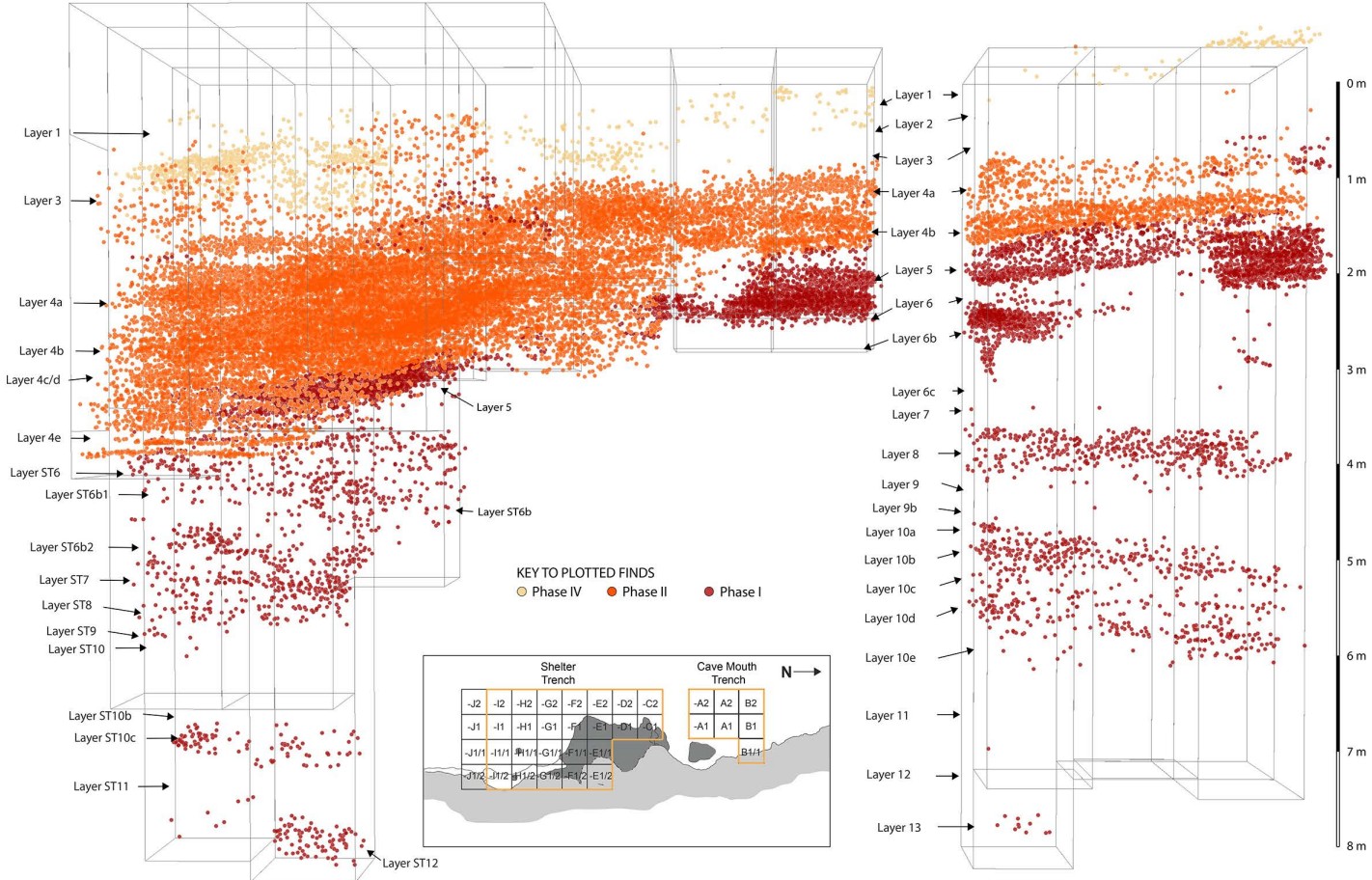

**Fig 15. Distribution of finds at Leang Bulu Bettue.** 3D-plotted positions of individual archaeological findings (stone artefacts, faunal remains, including shell, and other types) colour-coded according to the three major occupation phases in the main excavations: Phase IV (Austronesian-Neolithic), Phase II (~40 ka to 16 ka) and Phase I (>~40 ka).

teeth occur in layer ST12. The layer's average thickness is 40 cm, thinning towards the north to approximately 25 cm. The upper surface of this stratigraphic unit slopes more towards the north compared with the layers above it. The maximum depth excavated reached about 6.6 m below the cave floor (8 m BD).

Although layer 5 is laterally continuous across the excavated portions of the site, it is difficult to correlate the strata found below this layer in the Cave Mouth Trench and the Shelter Trench because they are separated by a distance of at least 6 m. Some inferences about stratigraphic correlations can be made, however: namely, that the deepest layer exposed in the Cave Mouth Trench (layer 13) is likely the same as layers ST10-10c in the Shelter Trench. This inference is based on similarities between the spatial distribution and types of finds in these deposits and the angle of slope of these layers, which is in close alignment. If so, layers ST11 and ST12 underlie layer 13 in the Cave Mouth Trench and are thus apparently older.

Overall, grain size and mineralogy do not vary in a systematic way through the stratigraphy of square A1, although detailed investigation is made difficult by the relatively few sediment samples analysed from this section. The most dramatic variation in the mean grain size is between squares A1 and -C1/-H1, the latter being considerably coarser grained. This pronounced increase in grain size is caused by the much higher abundance of limestone fragments in the sediments

from these parts of the site. It may reflect either an increase in local *in situ* decomposition of limestone and/or a decreased dilatation of these fragments in these sediments by predominantly volcanically sourced clays. Analysis of the Sr, Pb and Nd isotope composition of the windblow dust component of the sediments from square -C2 suggests a consistent source, with values consistent with the range reported for Central Sulawesi [76].

## Chronology of Phase I

As with the Phase II strata, no macroscopically visible carbonized plant material was recovered from the Phase I deposits. We also tried to extract microscopic pollen and organic residues for dating from layer 5 (Cave Mouth Trench) (see Supporting Information), but with no success. Shells were absent from these deposits, ruling out the use of $^{14}C$ dating. Therefore, to determine the age of the Phase I strata we undertook laser ablation U-series analyses of faunal remains from layers 5, 8, 10a, 10e and 11 in the Cave Mouth Trench. The dated fossil samples are shown in Figs 16,17. We also conducted luminescence dating of feldspar samples from layers 5, 9b and 10c in the Cave Mouth Trench, and layer 5 in the Shelter Trench. The latter required a novel approach to optical dating, one that we describe briefly in the Supporting Information and report in detail elsewhere [77].

Laser ablation U-series analyses of bovid teeth from the basal portions of layer 5 and overlying layer 4f (Cave Mouth Trench) suggest that the former accumulated at some stage between approximately 40 and 51.2–52.4 ka. This date is confirmed by two luminescence dates obtained on feldspar grains from layer 5 [77]. The first sample (LBB-II) was collected from square A1 (Cave Mouth Trench) at 205 cm below datum and at roughly the mid-point of this 50 cm-thick layer. The second sample (LBB15-OSL3) was collected around 3 m to the south from near the top of layer 5 in the east wall of square -C2 (Shelter Trench). Samples LBB-II and LBB15-OSL3 yielded within-error fading corrected ages of 33.5±9.1 ka (24.4–42.6 ka) and 40.1±6.2 ka (33.9–46.3 ka), respectively [66]. Together with the age of the U-series-dated bovid tooth from layer 4f, these dates suggest a maximum age of between 40 to 46.3 ka for the mid-to-upper part of layer 5. The U-series-dated bovid tooth from layer 5 provides an in-sequence minimum age of 51.2 ka for the lower part of this stratum.

In terms of strata underlying layer 5, we obtained a fading corrected luminescence age of 188.5±19.9 ka (168.6–208.4 ka) for feldspar grains collected from layer 9b in the Cave Mouth Trench (sample #LBB15-OSL2) [77]. Feldspars from underlying layer 10c in the same trench also produced a fading corrected age of 155.1±22.8 ka (132.3–177.9 ka) (sample #LBB15-OSL0) [77]. These two ages are within error of each other, allowing us to infer that layers 9b-10c fall broadly within the range of 132.3 and 208.4 ka [77]. Therefore, our results indicate that these deposits accumulated during the late Middle Pleistocene period.

Support for this chronology is provided by U-series analyses of fossil fauna from the deep deposits. U-series dating of a bovid molar (sample #3616) excavated from layer 10a (Fig 16J) yielded a minimum age of between 53.1±0.4 ka and 61.2±0.5 ka, while a proboscidean molar plate (lamella) fragment from underlying layer 10d (sample #LBB18) (Fig 16H) has a minimum age of between 40.3±3.8 ka and 81.1±2.4 ka. Below this level, a bovid molar from layer 11 (Fig 16F) yielded a stratigraphically consistent minimum age of between 67.4±3.0 ka and 72.2±2.4 ka. While it is important to remember that these are all minimum ages, it seems noteworthy that the U-series dates provided display a coherent pattern of increasing age with depth and do not contradict the luminescence chronology.

Minimum U-series ages for other fossils from layers 8, 10a, 10e, and 11 in the Cave Mouth Trench range between ~200–17 ka. The presence of age inversions and/or erroneously young estimates suggests these samples (specifically, samples LBB7, LBB10, and 3615) were reworked from older deposits. This hypothesis is confirmed by the distinct $^{234}U/^{238}U$ ratios of these particular samples, while others have clearly experienced recent U-uptake (i.e., overprinting). Hence, these samples were excluded from the analysis.

In sum, the uppermost part of the deep deposits (layer 5), and thus of the distinctive evidence for human occupation in Phase I, dates to between 40 and 51.2 ka. Despite the large uncertainties in the optical dating-derived dates [77], we may conclude from the luminescence ages for layers 9b-10c (Cave Mouth Trench) that the deepest stone artefact excavated

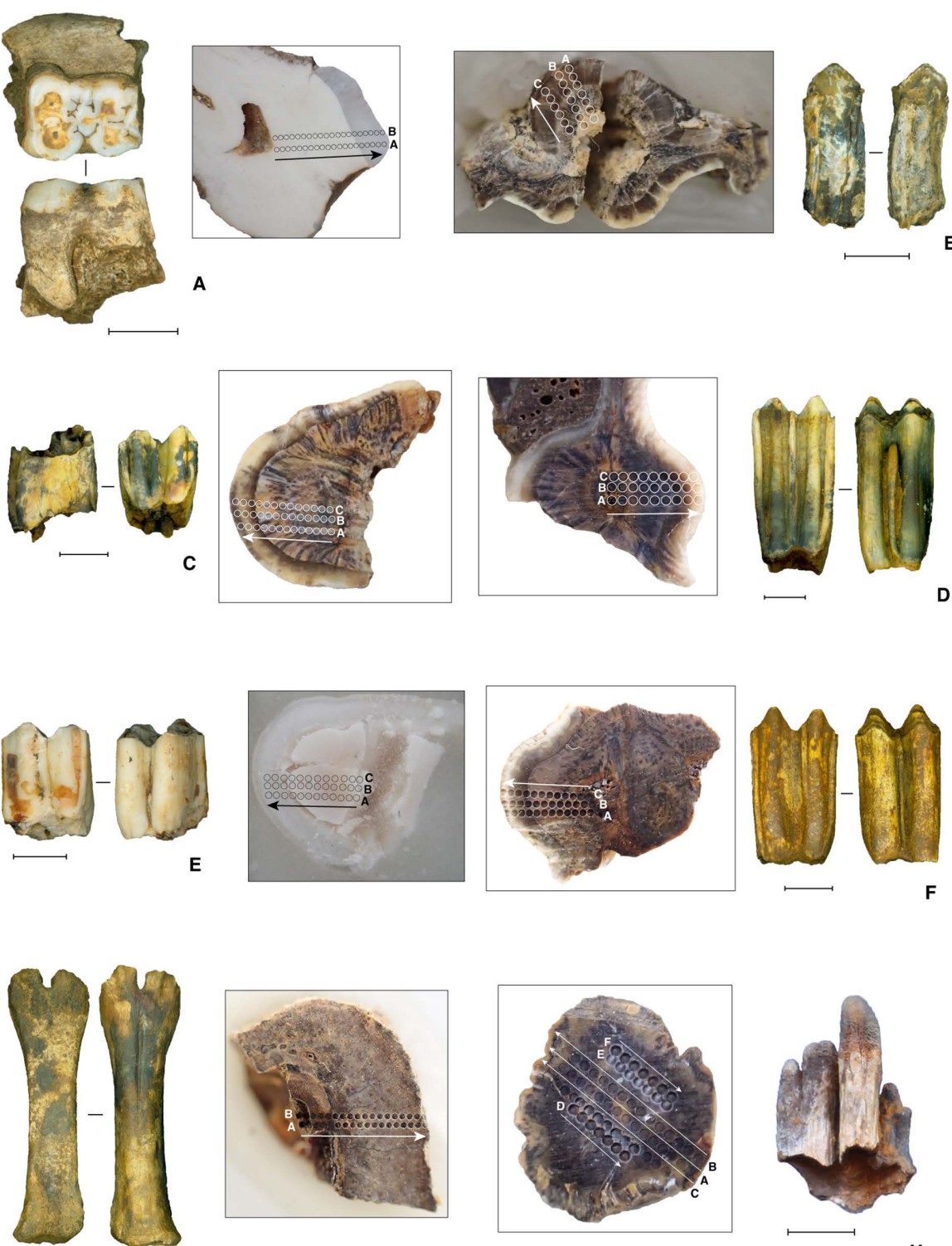

**Fig 16. Laser ablation U-series dated faunal remains, Leang Bulu Bettue.** Inset panels next to the faunal samples show close-up images of the laser spot profiles for each sample, annotated accordingly. **A**, Sample LBB-3 (layer 4a), suid molar embedded in mandible fragment; **B**, Sample LBB-7 (layer 5), anoa molar fragment; **C**, Sample LBB-10 (layer 8), anoa molar; **G**, Sample LBB-13 (layer 10), anoa molar; **E**, Sample LBB-15 (interface between layers 4b and 5), anoa molar; **F**, Sample LBB-17 (layer 11), anoa molar; **G**, Sample LBB-19 (layer 8), anoa metacarpus sinistral; **H**, Sample LBB-18 (layer 10d), *Elephantidae* lamel fragment. All scale bars are 10 mm in length.

thus far (layer 10a) dates to between 132.3–208.4 ka. It follows that the stratigraphically earliest evidence for human activity identified at the site up until now, cut marks on bones (see below) from the basal horizon in the Cave Mouth Trench (layer 13), is older than this date range, although by how much is presently unknown.

### Cultural and faunal remains in the Phase I deposits

Our understanding of the cultural and faunal records in the deep deposits at LBB is based principally on analyses of material excavated from the Cave Mouth Trench, although broadly the same findings occur below layer 5 in the Shelter Trench. The clear patterning in vertical distributions of lithic and faunal remains demonstrates the stratigraphic integrity of the deep deposits. As expected in such a complex depositional environment, there is evidence in the lower stratigraphy for sterile erosional cuts and small silt-filled depressions that seem to be natural drainage swales (e.g., layer 5), as well as more substantial structures that are probably infilled tree root cavities and soft-sediment deformation features (e.g., between layers 6b and 7 in the Cave Mouth Trench). Stone artefacts in the deep deposits are sparse, heavily patinated, and differ visually from the relatively fresh brown chert artefacts that dominate Phase II deposits. Furthermore, fossils from Phase I strata are typically in poor condition and often exhibit manganese-staining or are coated by calcrete and ferric (Fe) concretions. These specimens differ taphonomically from the much better-preserved and less heavily mineralised faunal elements in Phase II layers. These observations suggest there was minimal vertical displacement of artefacts (and fossils) below the Phase II deposits.

### Faunal assemblage in Phase 1

The vertical distribution of all faunal remains from squares -A1 and -A2 for which 3D coordinates were registered is shown in Fig 18 (see also Supporting Information) and the Number of Identified Specimens (NISP) and Minimum Number of Individuals (MNI) are quantified in S8 Table. Fig 18 excludes the material collected during wet sieving, cleaning of the baulks, and a few poorly preserved remains for which no 3D coordinates were recorded. Most Phase 1 faunal remains are concentrated in layer 5. Very few fossils were found in layer 8, although it yielded important proboscidean elements (see Supporting Information; S9 Table). The transitional interval between layers 9b and layer 10 and the underlying layers 10b and 10c contain a weak concentration of fossils. The lowest layer (layer 13) also contains few fossil remains, although it should be noted that excavation of this deposit was abandoned prematurely owing to groundwater intrusion. Layers 7, 9, 11 and 12 appear to be sterile.

Fig 18 summarizes the vertical distribution of taxa and abundances of the total number of fossil items per spit (for square -A1), the vertical distribution of "small" versus "large" vertebrates with depth, and the vertical distribution of burnt bone fragments and water-transported fossils. The major trends evident are the relative abundance of fossils in layer 5, and the preponderance of small-sized mammals in this layer, including *A. ursinus* and rodents. The main change in the sequence occurs at the boundary between layers 5 and 6. Below this transition, the frequency of faunal remains reduces drastically and the assemblage is overwhelmingly dominated by large-sized vertebrates. In addition, the number of burnt bones (already rare in layer 5 compared with the overlying Phase II strata) becomes extremely rare, and the quantity of water-rolled, rounded fossils increases below this transition. The latter trend supports the evidence based on the coarser-grained texture of layers 6–13, which indicates they were formed by fluvial processes. This depositional history would explain the relative rarity of small bone fragments, apart from reworked Oligocene fish teeth that seem to have behaved like sand grains and tend to be concentrated in the sandy layer 10 strata, especially near the base of this sandy unit (layers 10d and 10e). Since most of the reworked fish and crustacean remains were obtained from wet-sieving, they are not represented in Fig 17. Reworked fish remains were not included in the percentage calculations and thus do not inflate the "transported and water-rolled" percentages shown in the right-hand panel of Fig 18. Many of the large-sized vertebrate remains from layers 10 and 13 are water-rolled.

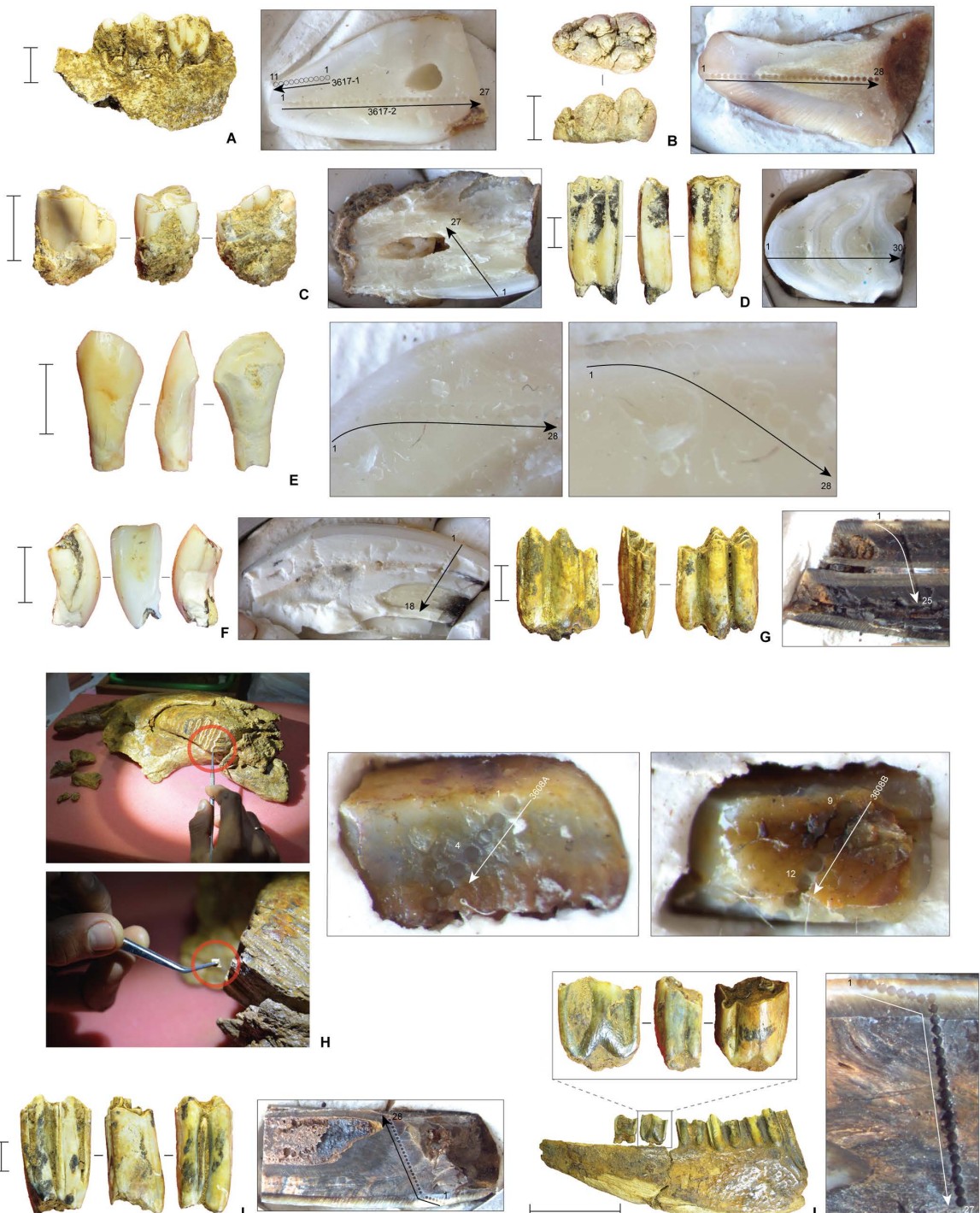

**Fig 17. Laser ablation U-series dated faunal remains, Leang Bulu Bettue.** Inset panels next to the faunal samples show close-up images of the laser spot profiles for each sample, annotated accordingly. **A**, Sample 3617−1 and 3617−2 (layer 4b), suid molar embedded in mandible; **B**, Sample 3610 (layer 4b), suid molar; **C**, Sample 3612 (layer 4f), bovid molar; **D**, Sample 3614 (layer 5), bovid molar; **E**, Sample 3613 (layer 5), bovid incisor; **F**, Sample 3609 (layer 5), bovid incisor; **G**, Sample 3611 (layer 8), bovid molar; **H**, Sample 3608A/B (layer 8), possible *Palaeoloxodon* cf. *namadicus* molar in mandibular ramus; **I**, Sample 3615 (layer 8), bovid molar; **J**, Sample 3616 (layer 10a), bovid molar in mandible fragment. All scale bars are 10 mm in length, with the exception of that in **J**, which is 50 mm in length.

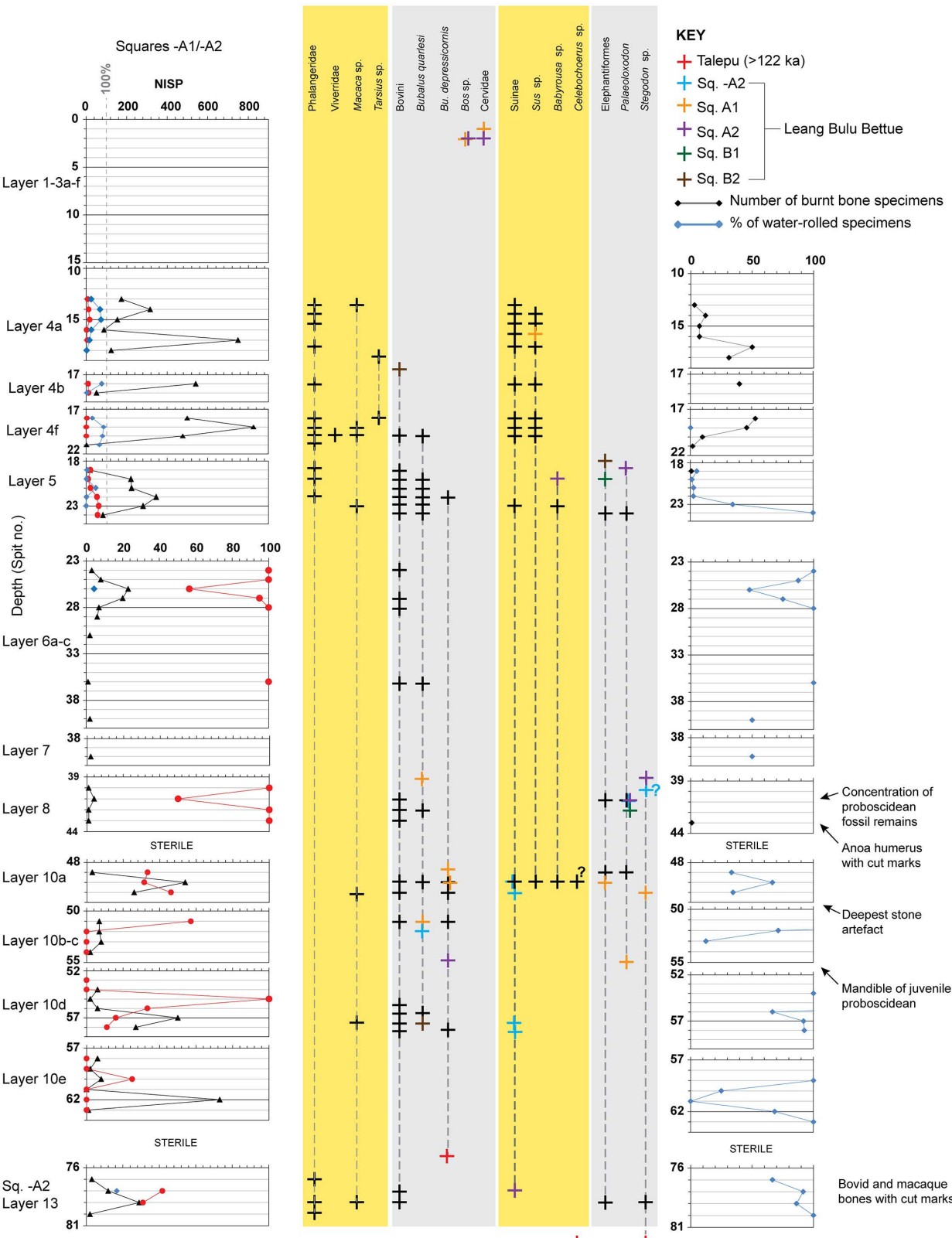

**Fig 18. Distribution, frequency, and characteristics of faunal remains at Leang Bulu Bettue.** Left panel: black triangles indicate the total number of vertebrate specimens per 10-cm-deep excavation spit (including fossil fragments obtained from wet-sieving, but excluding fish teeth); blue diamonds

indicate the amount of small-bodied vertebrate remains as a percentage of the total number of finds per spit (maximum 100%, including unidentifiable and identifiable bone fragments of Phalangeridae, primates, viverrids, birds, and reptiles); red diamonds indicate the amount of large-bodied vertebrate remains as a percentage of the total number of finds per spit (maximum 100%, including unidentifiable and identifiable bone fragments of Suinae, bovine, and proboscideans). Note the different scale for the upper part (layers 1-5) and lower part (layers 6a-13) of the panel. Central panel: vertical range chart of key taxa based on diagnostic faunal remains. Black crosses indicate the presence of a taxon in a particular spit of Quadrant -A1. Some ranges could be extended with identifiable remains from other quadrants. These are indicated with crosses of other colours (key in upper right corner). Right panel: black diamonds indicate the number of bone fragments with signs of heating/burning; blue diamonds indicate the amount of specimens with signs of water transport (rounding, water-rolled) as a percentage of the total amount of finds per spit.

The Phase I faunal assemblage comprises a mixture of modern and archaic fauna (Figs 19–20) (see also Supporting Information). The most common taxa are the two extant species of endemic dwarfed buffalo, lowland anoa (*B. depressi-cornis*, < 300 kg) and mountain anoa (*B. quarlesi*, < 150 kg) [78] (Fig 19). Fossils of suids (*Babyrousa* sp., *S. celebensis*, and possibly the extinct genus *Celebochoerus*) are present, but relatively uncommon. Further remains from the Phase I strata were assigned to macaques (*Macaca* sp.) and two now-extinct proboscideans (Fig 20). Broadly similar faunal remains were recovered from the Shelter Trench (Fig 21).

Elements attributed to smaller-sized taxa are less common (Supporting Information). A total of 31 rodent specimens was found in layer 5. Remains of the Sulawesi dwarf cuscus (*S. celebensis*) also occur in layer 5, although they are rare, and two dental elements attributed to Megachiroptera (flying fox) were recovered from this layer. The *Macaca* genus is represented by isolated dental elements in layer 5, an isolated upper first molar and a femur diaphysis in the layer 10 strata, and a macaque upper molar, a distal humerus and possibly an ulna diaphysis fragment in layer 13. Layer 13 also yielded eight vertebrae and vertebrae fragments of a large snake. Heavily built, and with a massive zygantrum, the vertebrae probably represent a member of the genus *Python* (Family Boidae Gray, 1825).

## Evidence for butchery

Direct evidence for butchery consists of cut marks on four bones from layer 13 (Fig 23A–23D), evincing a hominin presence at the deepest level exposed by our excavations in the Cave Mouth Trench (square A2). Notably, two of these specimens are from monkeys (*Macaca* sp.). A macaque left acetabulum with multiple cut marks and secondary trampling marks overlying was found in spit 77. The proximal edges of the cut marks show raised shoulders indicative of slicing (Fig 23A) [79–81]. Transverse microstriations run oblique and perpendicular across one cut mark; these relate to a secondary event, possibly trampling or other taphonomic activities [82,83]. An unidentified long bone fragment (probably *Macaca* sp. radius) with parallel cut marks underlying multiple fractures was found in spit 78 (Fig 23D). In addition to these specimens, an anoa rib fragment with a deep V-shaped stria was recovered from spit 79 (Fig 23C) and an unidentified bone fragment bearing multiple cut marks was found in spit 77 (Fig 23B). Higher in the stratigraphic sequence, we identified possible cut marks on two *Bubalus* sp. long bones from layers 8 and 10e (Cave Mouth Trench; Fig 23E–23F). In a more ambiguous association, a partial left mandibular ramus (Fig 20C) from a young, high-crowned adult elephant, possibly *Palaeoloxodon* cf. *namadicus* (Supporting Information), excavated from layer 8 (Cave Mouth Trench), was found near a flaked cobble tool (Fig 22).

## Stone technology in Phase I

Two contrasting technological approaches, or stone artefact "industries", were identified at LBB [57,84,85]. The Lower Industry consists of a total of 40 artefacts recovered from Phase I deposits in layers 5 of both main trenches and layers 6b, 8, and 10a in the Cave Mouth Trench (Figs 24–25), with artefacts from 2023 not included in the data analysis. The strata between 8 and 10a have not yielded definite stone artefacts, nor have the layers below layer 10a. The Upper Industry, identified in the Phase II deposits, consists of 25,748 artefacts excavated between 2013 and 2019 from the layer 4 strata across both the Cave Mouth Trench and the Shelter Trench. The reduction methods employed by the Upper

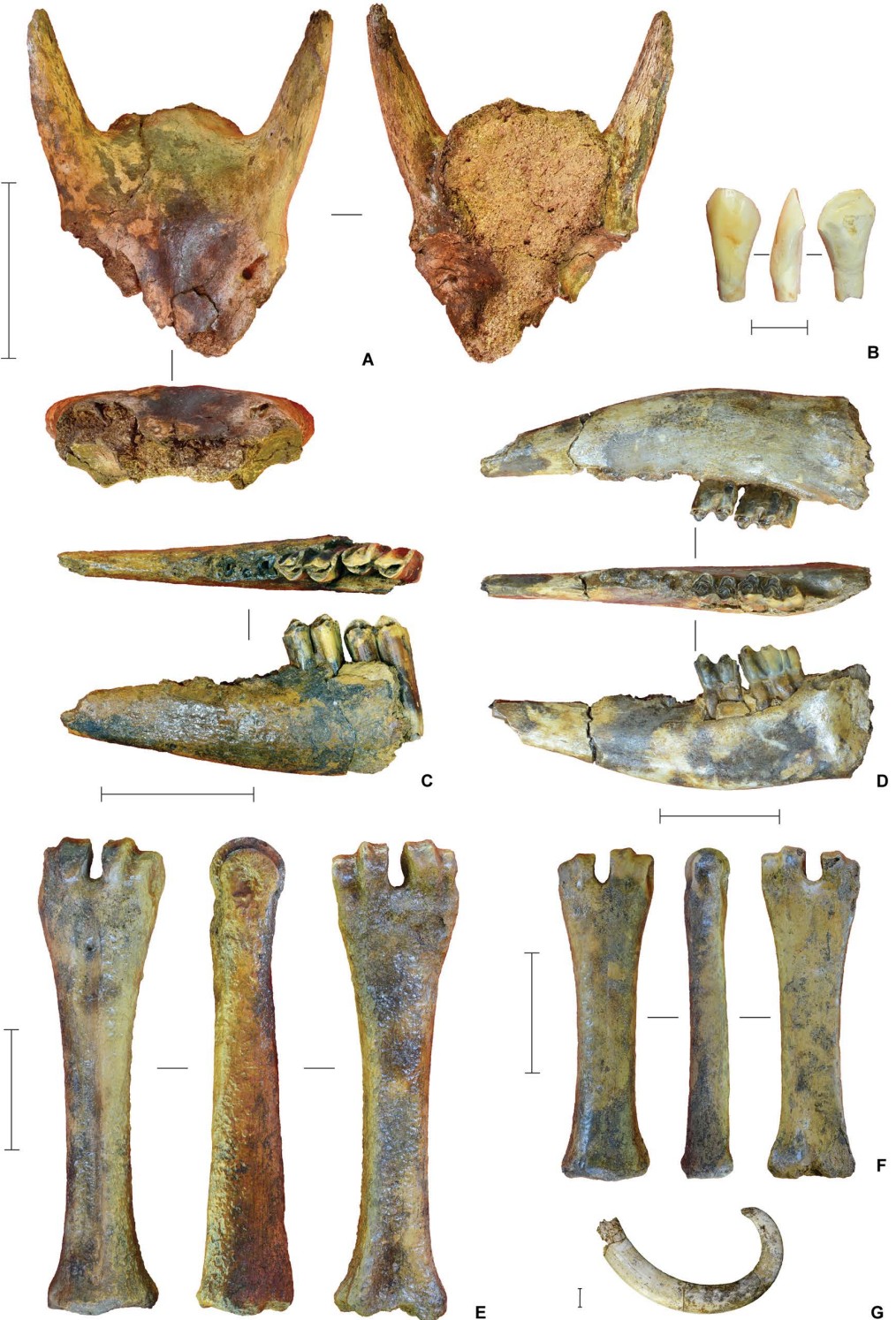

**Fig 19. Bovid fauna from Leang Bulu Bettue (Cave Mouth Trench). A**, Partial cranium of a small-sized anoa, probably mountain anoa (*Bubalus quarlesi*) (layer 10d; scale = 100 mm); **B**, anoa incisor (layer 5; scale = 10 mm); **C**, anoa mandible (layer 10b; scale = 50 mm); **D**, anoa mandible (layer 10c; scale = 50 mm); **E-F**, metapodials from lowland anoa (*B. depressicornis*) (**E**) and *B. quarlesi* (**F**); both specimens are from layer 10a (scale bars in **E** and **F** are 50 mm in length). **G**, babirusa left upper canine (layer 5). Scale = 10 mm.

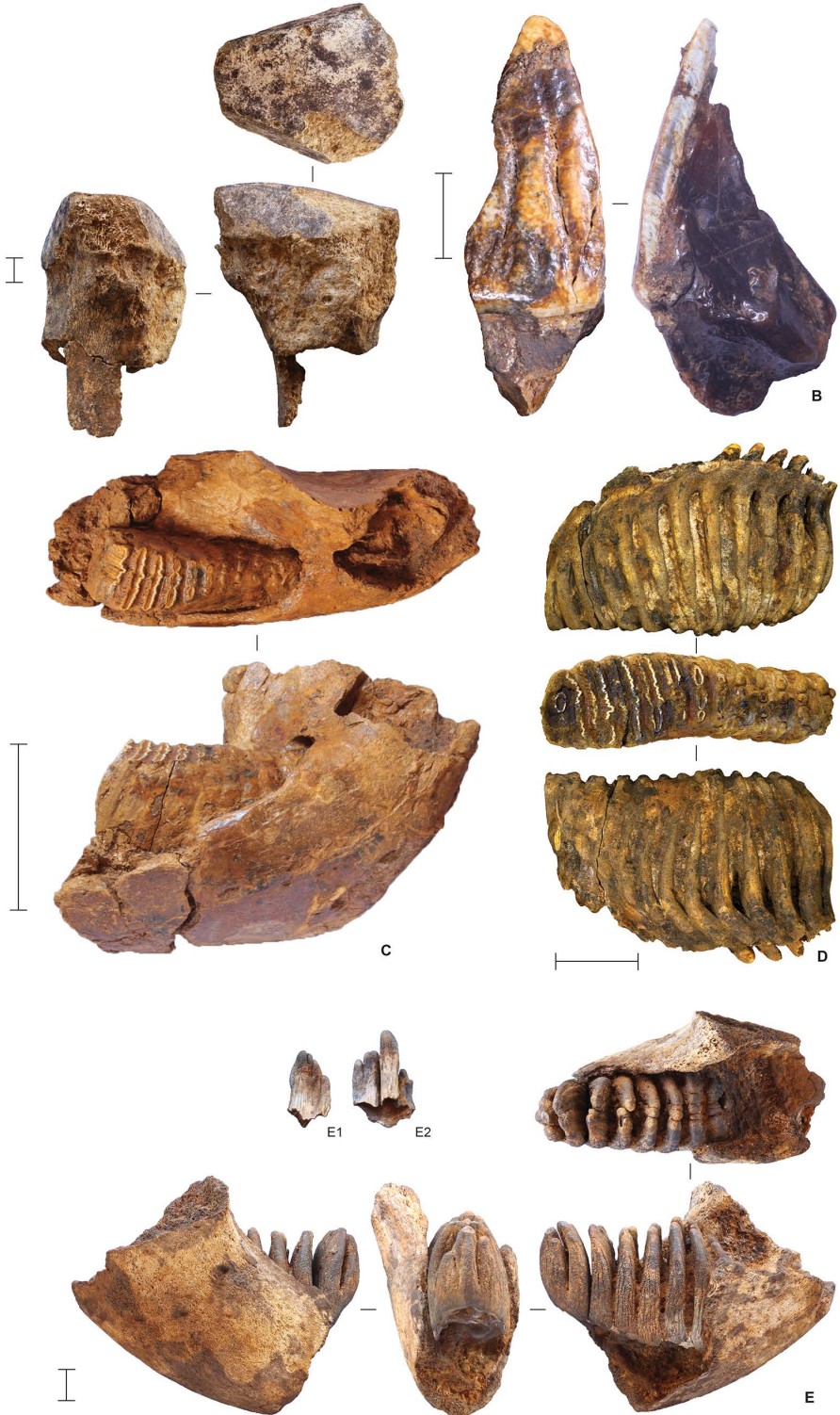

**Fig 20. Proboscidean fauna from Leang Bulu Bettue. A**, Partial metatarsal of an unidentified elephant from layer 5 (scale = 10 mm); **B**, *Stegodon* sp. molar fragment from the layer 10 fluvial sequence (scale = 10 mm); **C**, partial right mandibular ramus from a young adult elephant provisionally assigned to *Palaeoloxodon* cf. *namadicus* (layer 8; scale = 100 mm); **D**, isolated left lower molar from the previous individual **(C)** (layer 8; scale = 50 mm); **E**, juvenile elephant mandible from layer 10d (scale = 10 mm), with two isolated molar fragments from the same mandible illustrated alongside **(E1-2)**.

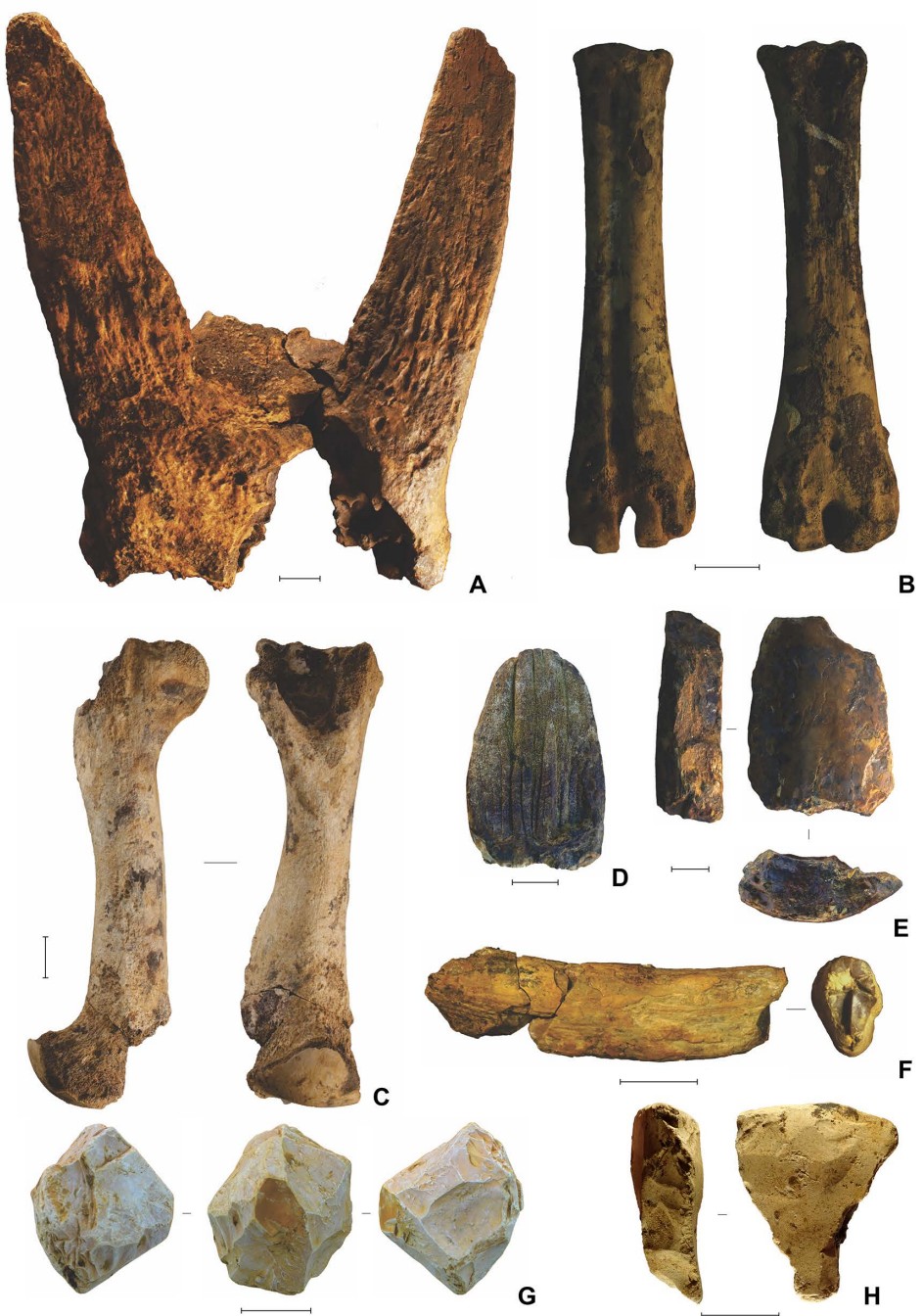

**Fig 21. Characteristic finds from the deep deposits in the Shelter Trench. A**, anoa frontal with horn cores (square -H1/1, spit 43, layer ST6b2); **B**, anoa metapodials (left: square -I1/1, spit 45, layer ST7; right: square -H1, spit 42, layer ST6b3/7); **C**, Suidae humerus (square -I1/1, spit 46, layer ST8); **D**, isolated proboscidean molar lamel, provisionally attributed to *Palaeoloxodon* cf. *namadicus* (square -I1/1, spit 31, layer ST6); **E**, proboscidean tusk fragments (square -I1/1, spit 47, layer ST8); **F**, possible *Celebochoerus* sp. tusk fragment (square -G1/1, spit 33, layer ST10); **G-H**, desilicated chert artefacts from slot trench; **G** radial core (square -I1/1, spit 33, layer ST6b3) and **H** retouched flake with 'perforator'-like morphology (square -C1/1, spit 40, layer ST8, 410-420 cm below datum). Scale bars 20 mm.

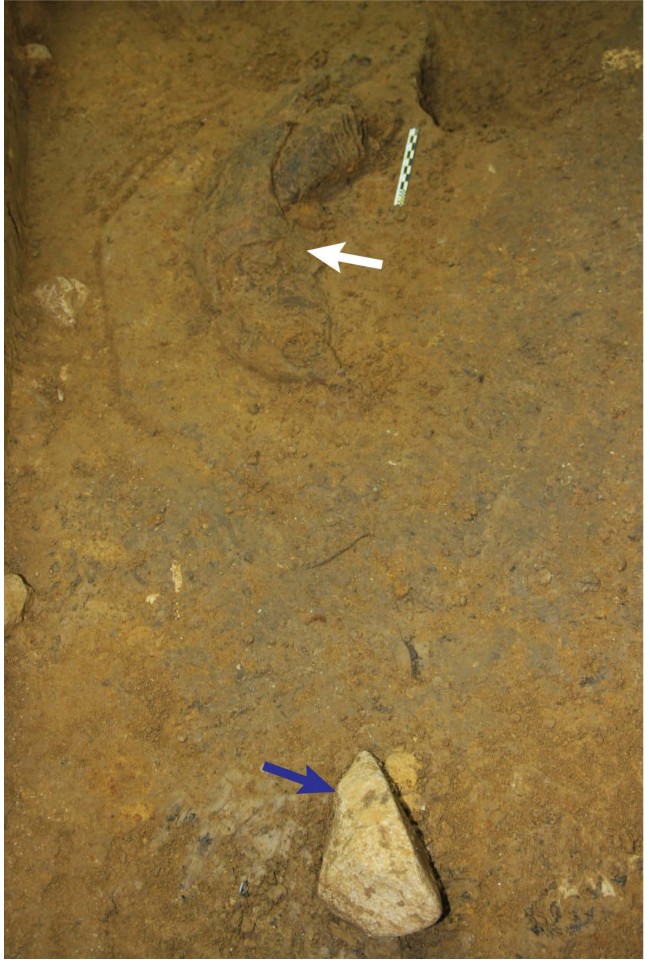

**Fig 22. Proboscidean faunal element and stone artefact in close proximity in layer 8 (Cave Mouth Trench).** The blue arrow indicates the stone artefact, a unifacially flaked 'pick' (see Fig 25A). The white arrow points to an associated right mandibular corpus of a proboscidean, possibly *Palaeoloxodon* cf. *namadicus*.

Industry stoneworkers have been summarised above and reported in detail elsewhere, along with a description of 13 Lower Industry artefacts from layer 5 [57,85]. Here, for the first time, we describe the Lower Industry artefacts from the deeper layers (6a, 6b, 8, and 10a), thus presenting our analysis of the Lower Industry as a whole.

The Lower Industry is characterised by cores made on river cobbles (Figs 24–25). Such cores were also found in the deep deposits in the Shelter Trench (Fig 21G–21H, Fig 26). The Lower Industry artefacts are made on volcanics, limestone, and chert. The chert artefacts have suffered 'white alteration' most likely from silica leaching from the damp, alkaline conditions of a limestone cave [see 57 p. 7 for a discussion of this phenomenon in the Maros-Pangkep karst systems], causing them to superficially resemble limestone. Local riverbeds are dominated by the locally-outcropping Tonasa Formation limestone and volcanic cobbles derived from the Camba Formation to the east, and water-rolled cobbles of these materials are available nearby. The cobble cores at LBB (2013−14 specimens from the Cave Mouth Trench) vary considerably in size (average 551±532.9 g), with the largest weighing nearly 2 kg (1859.3 g). Volcanic flakes were not discovered in association with these artefacts, and the cores may have been flaked outside the cave, though it is difficult to speculate from such a small sample size. Three cobbles were flaked unifacially on two margins to form a pick-like

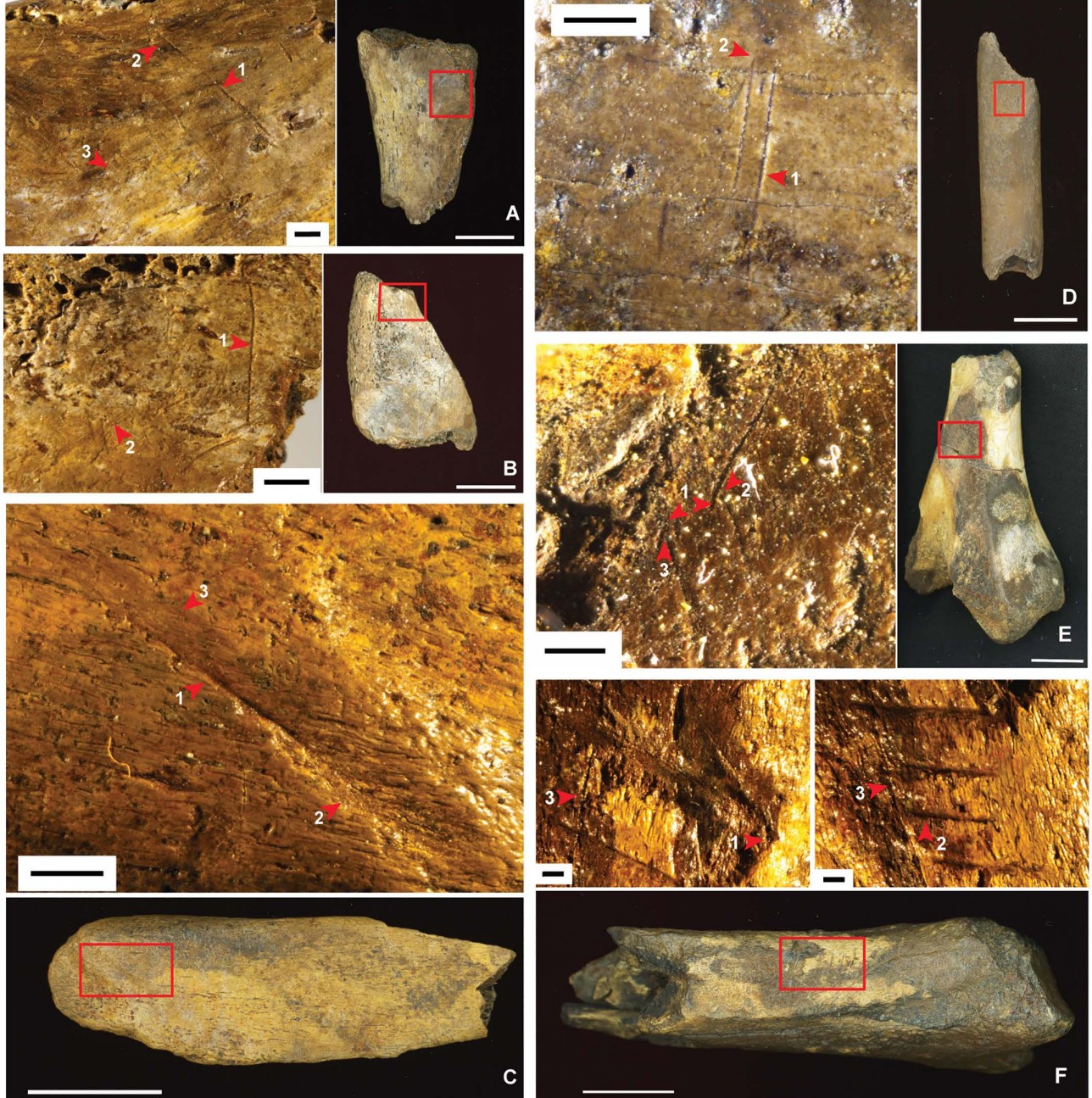

**Fig 23. Faunal remains with signs of butchery. A-D**, layer 13; **E**, layer 8; **F**, layer 10e. **A**, macaque left acetabulum (square -A2, layer 13, spit 77) with multiple cut marks. The proximal edges of the cut mark show raised shoulders (1) indicative of slicing. Initiation is identifiable due to a secondary groove running oblique to the main groove (2). Transverse microstriations (3) run oblique and perpendicular across the mark. The black scale bar in the micrograph is 1000 μm. The white scale bar in the right-hand image is 10 mm in length; **B**, unidentified bone fragment (square A2, layer 13, spit 77) bearing multiple cut-marks with patches of parallel micro-striations. The cut mark (1) is intruded into by natural striations (2) that are likely post-depositional in origin. The black scale bar in the micrograph is 1000 μm. The white scale bar in the right-hand image is 10 mm in length; **C**, bovid rib fragment (square A2, layer 13, spit 79) with a deep V-shaped stria. The cut mark comprises a deep, relatively straight stria that starts with a barb and terminates in a large, wide groove (1). The shoulder at the termination is exfoliated (2). The shape of the groove is V-shaped, where one wall is straight and the other flares out on an angle, making the mark appear wider (3). The black scale bar in the micrograph is 1000 μm. The white scale bar in the bottom image is 20 mm in length; **D**, unidentified long bone fragment (probably *Macaca* sp. radius) (square A2, layer 13, spit 78) with parallel cutmarks underlying multiple fractures. The specimen bears two symmetrical, V-shaped grooves with microstriations along the walls and bottom of the interior

(1). There are several fracture lines, as well as pitting and staining, that run transversely across both of the modifications (2), which denotes that the modifications were the primary event. The black scale bar in the micrograph is 1000 µm. The white scale bar in the right-hand image is 10 mm in length; **E**, anoa humerus with cutmarks inside a tuberosity, indicating intentional removal of muscle attachments. Several striae inside a large depression at the distal end of the humerus are evident (1). Several marks initiate parallel to the long-axis of the bone running proximal-distal before changing direction to run oblique to the long-axis (2, 3). The black scale bar in the micrograph is 1000 µm. The white scale bar in the right-hand image is 10 mm in length; **F**, anoa tibia showing five deep, parallel V-shaped striae running perpendicular to the long axis of the bone on the lateral margin of the right distal tibia. The large stria has markers that are consistent with movement of the wrist during butchery (1). The four distal striae are wide and slightly asymmetrical in morphology with a curved trajectory. The striae also have a strong shoulder, indicative of a sharp tool cutting into the bone at an angle (2). There is also evidence of long, thin fractures running over the top of the group indicating that the marks are older than the fractures (3). Black scale bars in the micrographs are 1000 µm. The white scale bar in the bottom image is 20 mm in length.

distal projection (Fig 24E and 24H; Fig 25A). Another cobble, from layer 5, was heavily flaked on one margin, possibly to create a pick-like projection with the opposite unmodified edge (Fig 24B and Fig 25C). Based on the sizes of flake scars, the flakes struck from the Lower Industry retouched cobbles were relatively small (average 32.1 ± 20.1 mm in maximum dimension), although large flakes (i.e., up to 109 mm in maximum dimension) were removed from one pick-like tool (Fig 24G). Among those not fully analysed are a small radial core recovered in 2023 that has bifacial centripetal flaking around the margin (Fig 21G) and a second small perforator-like retouched flake from square -C1 layer 8 (Fig 21H).

The non-pick cobble cores include an elongated limestone cobble with one scar on the end; a limestone multiplatform core with three flakes removed from two separate platforms; an elongated volcanic cobble with one end modified by three unifacial removals; and a volcanic cobble truncated by two unifacial removals. The Lower Industry includes three limestone flakes and 19 chert flakes, one of which is the largest chert flake recovered in the 2013 excavations, measuring 68.3 mm in maximum dimension and weighing 68.9 g. In addition, two modified chert flakes were recovered from these deposits, including one with two flakes removed towards the dorsal surface (Fig 24D), and a second with two unifacially retouched edges with a so-named perforator-like projection (see [35,86] for the definition of this term) between them (a 3D model of this artefact is available here: https://une.pedestal3d.com/r/qYFcntibU8). The latter artefact is desilicated and damaged and the original morphology is uncertain. Perforators share technological affinities with picks, with the latter being larger. Although recognised as elements of early hominin stone technology in the early Middle Pleistocene of Flores [87], perforators also occur in some modern human lithic toolkits in the region [86]. It should be noted that while the label perforator carries the connotation of function, is not intended to describe the manner in which this technologically distinctive artefact was actually used as this remains unknown.

The lowest recorded artefact is a large symmetrically flaked unifacial core-tool implement (Fig 24G–24H) that was made on a river-worn cobble of unidentified raw material, possibly fine-grained siltstone or volcanic (a 3D model of this artefact is available here: https://une.pedestal3d.com/r/SPiaqbVMjw). This artefact was excavated from the top of the layer 10 sequence (10a) in the Cave Mouth Trench at about 500 cm depth below surface (514 cm BD). It measures 193 mm in length, 138 mm in width and 82.8 mm in thickness. Nine conchoidal flake scars are evident, three on the left lateral margin and the rest on the right margin. The largest scar, on the right side of the implement, is at least 94.7 mm in length and is incomplete. Smaller retouch scars seem to have been present, but the object is so heavily weathered they are no longer discernible. The two margins converge to form a point with symmetrical edges. Technologically and morphologically, the artefact resembles a large Acheulean pick (see, e.g., [88–90]). The lack of any flakes in the layer 10 sequence adds weight to the argument that this was a tool brought to the cave rather than a biproduct of on-site reduction. The artefact is heavily weathered and was in extremely fragile condition when recovered, and was stabilised with B-76 Paraloid as part of the conservation treatment.

In sum, the Lower Industry can be characterised as a relatively straightforward cobble-based core-and-flake technology, compared to the Upper Industry. As such it is broadly similar in character to the "Lower Industry" at nearby Leang Burung 2, dated to earlier than ~50 ka [54], the Talepu lithic technology dated to 194–118 ka [43], and the Calio lithics

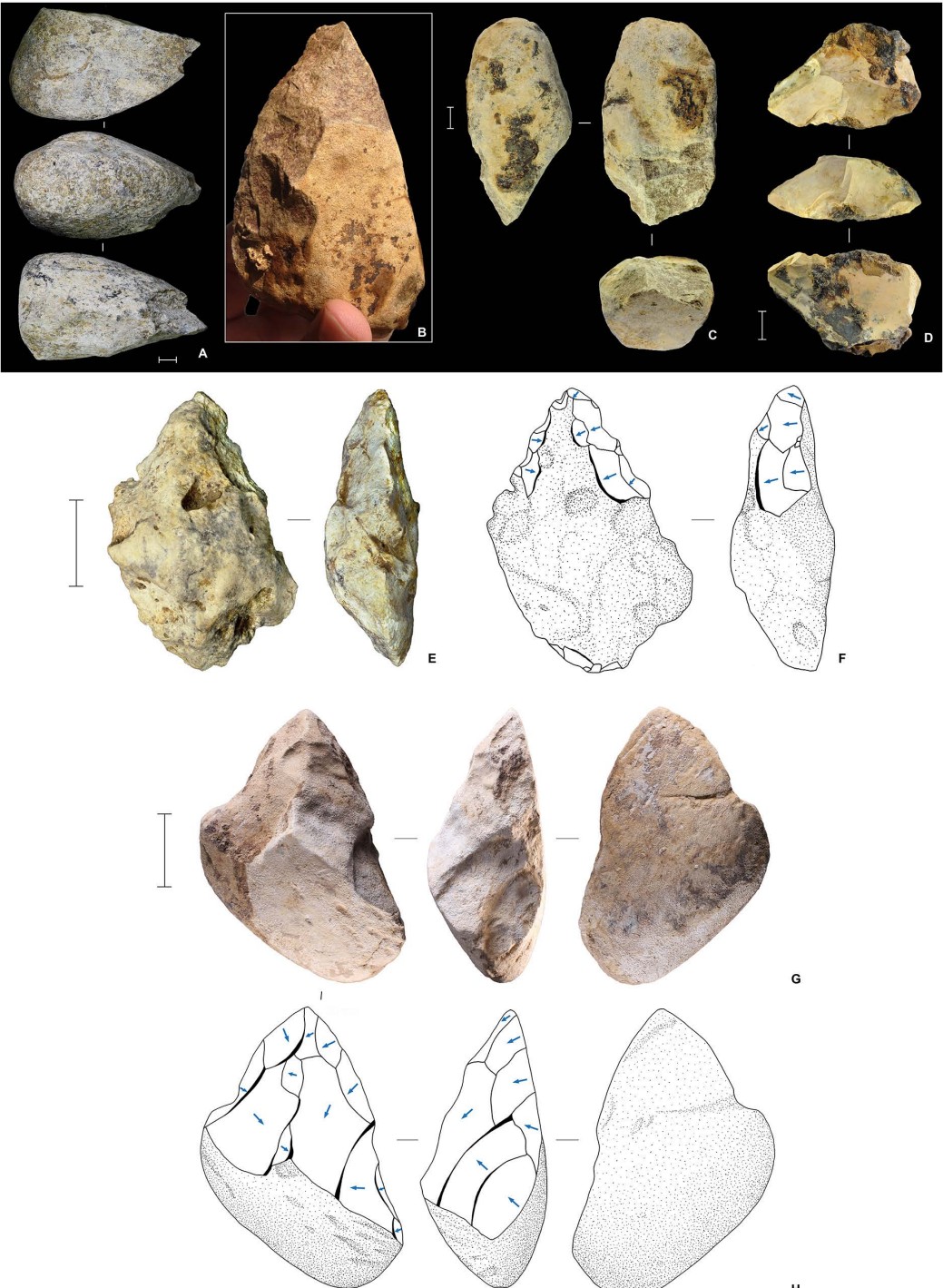

**Fig 24. Lower Industry stone artefacts from LBB. A**, Retouched volcanic flake, layer 5 (scale = 10 mm); **B**, unifacially retouched volcanic pick, layer 5; **C**, bifacially retouched volcanic cobble, layer 5 (scale = 10 mm) (a 3D model of this artefact is available at: https://une.pedestal3d.com/r/NnRSE6oPqd); **D**, retouched chert flake, layer 8 (scale = 10 mm); **E**, unifacially retouched limestone pick (layer 5), with diacritical illustration **(F)** (scale = 50 mm); **G**, unifacially retouched volcanic pick (layer 10a), with diacritical illustration **(H)** (scale = 50 mm) (a 3D model of this artefact is available at: https://une.pedestal3d.com/r/SPiaqbVMjw).

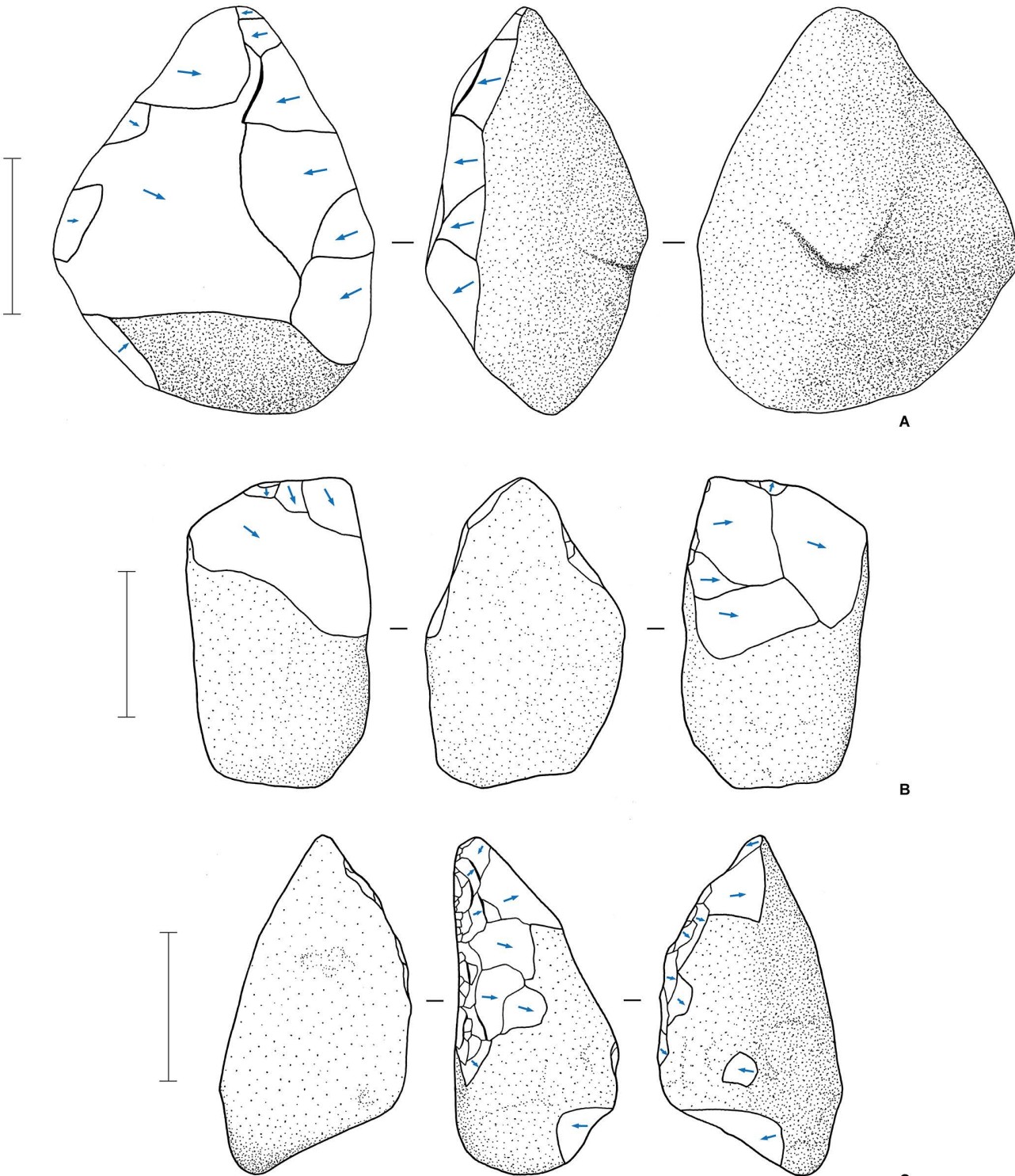

**Fig 25. Picks and pick-like core tools from the Lower Industry at Leang Bulu Bettue.** All three specimens were made on volcanic cobbles. **A**, layer 8; **B**, layer 8; **C**, layer 5. The pick in **A** was found adjacent to the right mandibular corpus of a proboscidean, possibly *Palaeoloxodon* cf. *namadicus* (see Fig 21). Scale bars 50 mm.

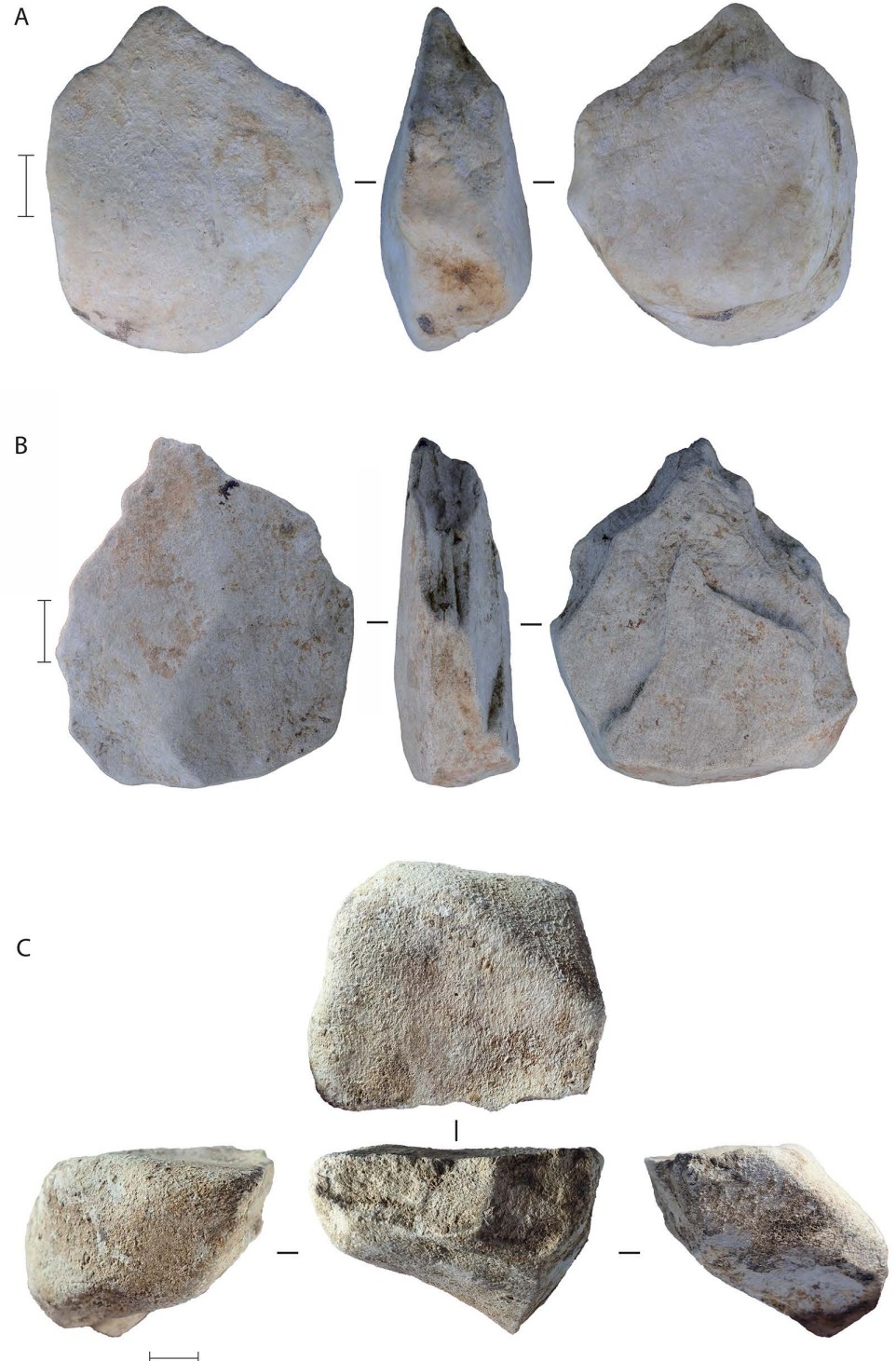

**Fig 26. Limestone cores from the deep deposit in the Shelter Trench. A** is from layer ST6b, **B** layer 4c/5, and **C** layer ST7. Scale bars = 20 mm.

dated to 1.04 Ma [44] (see also Supporting Information). To our knowledge, pick-like tools have only been recovered in undated surface contexts at Talepu and other open-air sites in the Walanae Depression. However, a morphologically similar pick-like tool was excavated from the lower stratigraphy in Leang Burung 2 (square D10, spit 55, layer A) [54]. The layer from which this Leang Burung 2 artefact was excavated has a maximum age of ~50–46 ka [54] (see also Supporting Information). Broadly similar pick-like artefacts were excavated from Wolo Sege in Flores, dating to >1.02±0.10 Ma [91]. It is unclear whether picks represent a purpose-made tool in Indonesia, although they are typologically similar to Acheulean tools thought by many scholars to have been intentionally designed forms made according to distinct mental templates [88–90] (but see [92,93] for a critique of this notion).

## Discussion

This paper provides the first complete description of excavations at the Pleistocene site of LBB. Prior publications [56–60] have described the upper cultural layers (4a-f) at LBB to a depth of around 2 m below the ground surface of the cave, representing a key time period in modern human history between approximately 40 ka to 16 ka. In the present study, we refer to this period as the Phase II occupation and report for the first time archaeological findings excavated from the underlying strata—Phase I occupation—to a depth of about 8 m, at which point excavations were terminated at the water-table without reaching bedrock or culturally-sterile layers. This lower stratigraphy, ranging from layers 5–13 in the Cave Mouth Trench and layers 5 to ST12 in the Shelter Trench, exhibits characteristics both in terms of layer composition and finds that differ strikingly to the overlying layers (4a-f), the earliest of which post-dates ~40 ka. The lowest stone artefacts at LBB have been dated to 132.3–208.4 ka

In summary, the major trends evident in the Phase I and II faunal assemblages at LBB are the relative abundance of fossils in layers 4a-f and layer 5, dominated by small-sized mammals. In the layer 4 sequence (early Phase II), there is a relatively high frequency of burnt bone fragments, which are largely absent (though not completely lacking) from layer 5 and below (Phase I). In terms of long-term trends in megafauna exploitation, Phase II is dominated by suids and the earlier Phase I by anoas; specifically, bovinae including anoa dominate the megafauna assemblage of Phase II, often constituting 100% of the non-fish vertebrate remains. Anoas appear to have been extirpated from lowland regions of southern Sulawesi, or else their populations were reduced considerably in numbers, by 30 ka [94]. Notably, at Goa Topogaro 2 in Central Sulawesi, anoa remains were most common in the deepest layers, dating from 30 kyr cal BP to at least 42 kyr cal BP [22].

In addition to anoa, smaller-bodied species are also represented in Phase I. Cuscus remains occur in layer 5, and two of the four bones with apparent butchery traces (cut marks) identified in the deepest stratum encountered thus far (layer 13 in the Cave Mouth Trench) are from *Macaca* sp. of monkeys. Our understanding of the subsistence strategies used by the Phase I occupants, including the frequency with which such small-sized, arboreal/semi-arboreal prey species were exploited, and the hunting methods employed, is limited. However, purposeful hunting of primates is often implicitly associated with modern humans as it is presumed to require the use of traps and/or mechanical projectile weaponry [95–97]. The presence of lithic artefacts also demonstrate human activity in the cave during Phase I.

To our knowledge, layer 5 bears the chronologically youngest record of proboscideans to our knowledge documented in Sulawesi, although it is unclear if the presence of these fossil remains relates directly to human activities. The partial left mandibular ramus from layer 8 (Fig 20C) assigned to *P.* cf. *namadicus* (Supporting Information S6-S11 Table) is the most complete fossil specimen of an archaic elephant presently known from Sulawesi. No evidence for proboscideans was found in Phase II deposits. Up until now the known fossil record of these extinct faunas on Sulawesi had been confined largely to coarsely-dated finds from open sites in the Walanae Depression, so excavating proboscidean fossils from a securely stratified and dated context in Maros-Pangkep is noteworthy [43].

Phase II yielded rich cultural remains and abundant finds that are suggestive of much more intensive human occupation of the site than in the preceding Phase I. The dense faunal assemblages in layers 4a-f are characterised by shells

of the freshwater snail *T. perfecta*, and fragmented and often burnt elements of bear cuscuses (*A. ursinus*), rodents, macaques, and other small mammals. Remains of birds, reptiles, crabs, and fish are also present. In contrast to Phase I, the largest animal represented in significant proportions is the endemic, small-bodied wild suid *S. celebensis*. Figurative depictions of animals in the dated Late Pleistocene rock art of Maros-Pangkep are dominated by representations of *S. celebensis* [31,98], though anoa are also depicted. The earliest known surviving image of the former animal, a large painting at Leang Karampuang, has a minimum age of 51.2 ka [25]. Rock art depictions of *S. celebensis* date to at least 48 ka at Leang Bulu' Sipong 4 [25] and 45.5 ka at Leang Tedongnge [31].

The Phase II lithic technology, dubbed the Upper Industry [57], is distinct from the Lower Industry of Phase I, showing more diverse technological variation and smaller average artefact size, as well as relatively higher use of imported silicious chert, evidence for burning, and a greater number of apparently imported raw materials in general ( [57]; see [85] for an in-depth analysis of the technological changes at the site; and Supporting Information). This Upper Industry technology is generally consistent with approaches to stone reduction employed by the earliest groups of *H. sapiens* in southern Wallacea at an equivalent time period (~44–30 ka until post-LGM), including at Liang Bua [86], Makpan, Laili, Asitau Kuru (formerly known as Jerimalai) and Matja Kuru 2 [99], which suggest a widespread trend towards greater focus on exploitation of cryptocrystalline raw lithic materials, especially chert. The trend towards smaller average artefact size could be a direct result of this. Some describe a general shift to smaller artefacts as a cultural phenomenon of "miniaturization" [e.g., 100]; however it is worth observing that this may simply be a by-product of other behavioural changes rather than a direct choice, at least at LBB. Here, higher quality raw material was carried over greater distances during the Phase II period of activity, often in the form of large flake blanks [56,85], which tend to be smaller than immediately available river cobbles, and more of the reduction process occurred within the cave. These factors lend themselves to more heavy reduction of smaller blanks, and consequentially the occurrence of smaller flake and core artefacts, on average, than seen in the earlier deposits. By contrast, the Lower Industry at LBB is more similar in character to the late Middle Pleistocene core-and-flake technology at Talepu [43], with both characterized by an uncomplicated, low-effort approach to reducing local cobbles [101]. Similarly, the higher proportions of burning damage (i.e., potlid scars and other signs of heat breakage) [e.g., 72] is most likely a secondary product of increase in fires at the site.

The abrupt change in the archaeological sequence at LBB between the Lower and Upper Industry (Phases I and II respectively) seems to indicate that a shift in human behaviour had taken place by ~40 ka, in a similar pattern to that seen at Leang Burung 2 [57]. First excavated in 1975, Leang Burung 2 was at that time the oldest and best-documented Pleistocene locality in Sulawesi [52]. A renewed program of excavations revealed that this rock-shelter has a complex depositional history, with at least one major phase of erosion and re-infilling [54]. Despite these problems, recent work shows that two distinct phases of Pleistocene human occupation took place at Leang Burung 2 [54]. The earliest phase, evident in Layers I and A-C, is characterized by a lithic technology (also dubbed the Lower Industry) reminisce of the Talepu lithics, that was largely focused on the use of local limestone and volcanic cobbles to manufacture large core-and-flake tools (see Supporting Information), as well as the exploitation of endemic megafauna, specifically anoas and suids. Excavations also yielded proboscidean molar fragments from the topmost layer attributed to this early occupation phase (Layer I), a massive weathered soil horizon. The time-depth of the early phase with its Talepu-like industry and extinct megafauna is poorly constrained, and it is uncertain whether deeper cultural materials are present as excavations were abandoned owing to groundwater intrusion [54]. The upper occupation phase at Leang Burung 2, in contrast, is characterized by smaller chert tools (including macroblades and edge-glossed flakes) that are the outcome of a lithic reduction strategy that was relatively complex compared with the earlier technology, pigment processing, and intensive exploitation of freshwater shellfish and small-sized vertebrates such as cuscus, monkey, birds, bats and civets. There is no evidence of proboscideans or other extinct megafauna in the rich faunal assemblages recovered from these deposits. This upper occupation phase at Leang Burung 2 bears some of the hallmarks of uniquely modern human behaviour [58–60] (but for critiques of this concept, see [102,103]).

To summarise, the occupation sequence at LBB closely mirrors that at nearby Leang Burung 2, with its evidence for a major cultural and subsistence shift in pre-LGM deposits. Discernible in the archaeological record at both sites is a bipartite division of occupation into a lower phase—Phase I—characterized by a fairly straightforward stone technology, exploitation of large-bodied game, and the presence of archaic fauna, and an upper phase—Phase II—that is distinguished by a richer array of human behaviour, including a relatively more diverse lithic toolkit, pigment use, and a broad-spectrum subsistence economy characterised by intensive exploitation of shellfish resources and small-bodied vertebrates. Skeletal remains of a *H. sapiens* individual confirm that this species utilised the cave [56]. Dating evidence from LBB puts the transition between these two occupation phases at around 40 ka, while the less precise chronology from Leang Burung 2 [54]—constrains the Phase I to II transition to between roughly 50 ka and 35 ka.

## Causes of the behavioural shift

Importantly, this change in site activities, now evident at the two oldest and most deeply-excavated sites in Maros-Pangkep, does not seem to be linked to the effects of a change in environment and/or climatic conditions. High-resolution palaeoclimate records derived from lake sediment cores in Central Sulawesi indicate a very wet climate and closed-canopy rainforest during much of Marine Isotope Stage 3 (MIS3) (~58–30 ka) [104–110]. This time was followed by the abrupt onset of a period of severe drying in the transition to the LGM, with the arid phase documented from ~33–16 ka at Lake Towuti, and from ~29–14 ka at nearby Lake Matano [104–107]. Biomarker records ($\delta^{13}C_{WAX}$) from these two lakes suggest grassland and open-canopy forest expanded during a more arid and seasonal MIS2 [108], with substantial expansion of $C_4$ grasslands between 30 and 18 ka [104]. A return to wetter and more humid conditions is evident by $14.8 \pm 0.38$ kyr BP during the transition to the interglacial, with the $\delta^{13}C_{WAX}$ reaching its Holocene average at $11 \pm 0.28$ kyr BP [104]. Speleothem $\delta^{18}O$ records obtained from U/Th analysis of 77 stalagmites in the Maros-Pangkep karsts lack precise coverage for the time interval of interest [109]. Another study [110] suggests that textural laminae in a single stalagmite from Maros-Pangkep (Leang Saripa) reflect periods of rapid environmental change that took place in the area at around 44.7 ka and 22.5 ka, possibly due to volcanic eruptions. However, trace element and C-O-Sr isotope analysis of a stalagmite from Bumi Cave with a similar growth history revealed no such evidence for volcanic activity [110].

In sum, based on the available palaeoclimate records from Sulawesi it is likely that lowland rainforest was the dominant biome in the Maros-Pangkep karsts throughout MIS3. The records available are not extensive, it should be emphasised; however, current data suggests there is no evidence for a major environmental change around 40 ka that might explain the behavioural disconformity evident in the archaeological record of this region. As the earliest artefacts at LBB date to 132.3–208.4 ka we can assume they were not made by *H. sapiens* but rather by an earlier species that was presumably also responsible for the stone artefacts from Talepu (194 ka) and Calio (>1.04 Ma), so when and how did the transition occur? We propose two possible explanations for the shift. Both scenarios assume that the rock art of the region was produced by *H. sapiens* as this is more parsimonious than attributing art to an archaic hominin, thus providing a minimum arrival of ~51.2 ka for the species.

The first possibility is that the entirety of Phase I reflects the last period of a long history of occupation by a group of archaic hominins–as already noted, likely those responsible for the early lithic artefacts recovered from the Walanae Basin (Talepu and Calio) ~80 km to the northeast – that was replaced by an incoming group of modern humans around 40ka, with the arrival of the latter being the cause of the change in the LBB record (i.e., the onset of Phase II). The archaic hominins, based on the present state of knowledge from the wider region, could have been *H. erectus* and/or a taxon closely related to *H. floresiensis*, Denisovans, or an as-yet undocumented hominin species that is now extinct [43,44]. This shift is not seen in the rock art, however, with sites dating to around 50 ka and no significant stylistic changes until the Holocene [63]. The proposed reason for this is that there was a sustained overlap between both species, perhaps with *H. sapiens* initially limited to coastal zones where they could readily exploit the abundant supply of protein-rich marine resources (especially shellfish), as they were doing in Timor (but see [111]) and elsewhere in southernmost Wallacea

([112]; see also [113]) and in line with the popular 'Coastal Colonisation' theory for human dispersal [e.g., 114]. This speculation is difficult to test as any early coastal sites, if they existed, would be on the now-drowned Spermonde shelf.

A second possibility we propose is that the earliest *H. sapiens* in the region produced lithic technologies that more closely resembled the technology of the archaic hominins of the Walanae basin at least 200 ka to 1.04 Ma [43,44] than those made by later *H. sapiens*, and that the technological disconnect cannot be described by a species-level replacement. The apparent technological continuity may be the result of contact between two groups, or simply a convergent response to the same resources and conditions. This situation is in keeping with the evidence for the behavioural flexibility of early modern humans as they spread out of Africa and began to colonise unfamiliar environments [102]. The technological and faunal shift at around 40 ka, resulting in Phase II and the lithic Upper Industry, would therefore reflect an unknown local trigger, spontaneous innovation, and/or the arrival of a second wave of *H. sapiens*. One possibility is that intensive exploitation of anoas by the founding group led to over-hunting of this solitary forest-dwelling bovid, and its subsequent localised extirpation, necessitating a switch in procurement strategies to a more diverse range of smaller prey, including *S. celebensis*. This change in the group's hunting culture may have had a flow-on effect on its material technology, including the development of the distinctive lithic toolkit of Phase II.

## Conclusion

The recent discovery of stone tools dated to >194 ka at Talepu, > 1 Ma at Calio in the Walanae Basin, and now 132.3–208.4 ka at LBB, has shown that archaic hominins of as yet unknown taxonomy had established a presence on Sulawesi by at least the late Early Pleistocene [43,44]. The identity and origin of the original inhabitants of this large Wallacean island are no less enigmatic than the later history of this group and the final phase of its occupation, perhaps represented at LBB. The first modern humans (*H. sapiens*) to reach Sulawesi may have come ashore by at least 50 ka [17–18], and potentially as early as 65 ka [19], if they took the northern route to Australia as we are led to believe by recent studies [15].

This species shift is reminiscent of events documented in the early prehistory of other parts of the globe. In western Eurasia, for example, *H. sapiens* arrived by ~45–43 ka, and by five millennia later the last-surviving Neanderthals were extinct [115,116]. On Flores, the last evidence for *H. floresiensis* in the fossil record is 60–50 ka [36,117], suggesting this endemic island species disappeared not long after modern humans entered Wallacea. It is unclear if the two hominin species of Sulawesi overlapped, and we offer two alternate and competing scenarios for the change and continuity represented at the site of LBB.

At present, LBB is the only archaeological site on Sulawesi that has yielded a deeply stratified archaeological deposit with dates that fall broadly within the relevant timeframe. Our deep-trench excavations at this locality since 2013 have yielded archaeological evidence for a distinctive early human occupation phase (Phase I) that dates from the late Middle Pleistocene in the deepest part of the deposit uncovered thus far. These Phase I inhabitants used a low complexity core-and-flake lithic technology (the Lower Industry) and appear to have exploited a range of endemic land mammals, in particular anoas, but including at least some smaller-sized animals such as monkeys and cuscuses. There is no evidence for pigment use or other hallmarks of symbolic behaviour in the Phase I deposits. The uppermost layer attributed to the distinctive Phase I occupation (layer 5) dates to between approximately 51.2 ka and 40 ka, thus representing either a late-surviving archaic species or a level of cultural continuity with their successors, *H. sapiens*. A novel lithic toolkit thereafter appears at LBB in the succeeding Phase II strata (layers 4a-f), in concert with a diverse suite of pronounced cultural changes at this particular site, including a shift in the subsistence economy and the first appearance of symbolic behaviour. Broadly the same pattern is evident at nearby Leang Burung 2, where there is evidence for a similar change in the faunal and lithic sequences [54].

The deepest and oldest archaeological evidence for the Phase I hominin occupation at LBB pre-dates the earliest established presence of modern humans not just in Island Southeast Asia, but anywhere outside Africa. It should also

be noted that our deepest trench at LBB has not yet exposed the limestone bedrock or culturally-sterile deposits, suggesting the potential for discovering older cultural layers at this particular site. The available evidence suggests there is a time gap of at least 11,200 years between the first evidence for rock art production in the Maros-Pangkep karst area (minimum of 51.2 ka at Leang Karampuang; [25]) and the material shift in the archaeological record at LBB and Leang Burung 2, including the first indications (~40 ka) of the presence of ochre in the deposit at the former site ~40 ka [58].

In light of these findings, the implications of the transition between Phases I and II at LBB (and Leang Burung 2)—the behavioural disconformity—are unclear, although we have considered two possible scenarios that are consistent with the evidence available to us. It is not possible to test either of these hypotheses until we have considerably advanced our knowledge of the archaic hominins of Sulawesi, and, most importantly, until diagnostic fossils from the earliest inhabitants have been discovered—in particular below the behavioural disconformity at LBB. Quite apart from the taxonomic affinity of these early hominins, which is completely unknown, very little is understood about the history of their occupation, including: when the founding population reached Sulawesi and the length of time it persisted on the island; the geographical distribution and population densities of these hominins; the variability of their material technology, diet, and local behavioural adaptations to the diverse insular ecosystems (e.g., ecological preferences) and endemic faunal communities of Sulawesi; how they coped with climatic instability and the long history of wet/dry cycles on Sulawesi; and when and how they became extinct. We anticipate that future research in Maros-Pangkep and other parts of Sulawesi, such as the Walanae Depression as well as areas beyond the southwestern peninsula where research has historically been focused, will throw new light on this little-known aspect of the early human story.

## Supporting information

**S1 File.**
(DOCX)

**S1 Table. X-Ray Diffraction (XRD) results for Leang Bulu Bettue sediment samples.**
(PDF)

**S2 Table. Isotope geochemistry analysis results for Leang Bulu Bettue sediment samples.**
(PDF)

**S3 Table. U-series results on bones and teeth from Leang Bulu Bettue.** No age calculations were carried out for U concentrations of ≤0.5 ppm or U/Th ≤ 300, but not negative (indicated in red). Negative U/Th are due to the average background being higher than the specific measurement. All errors are 2σ.
(PDF)

**S4 Table. U-series results on bones and teeth from Leang Bulu Bettue.** No age calculations were carried out for U concentrations of ≤0.5 ppm or U/Th ≤ 300, but not negative (indicated in red). Negative U/Th are due to the average background being higher than the specific measurement. All errors are 2σ.
(PDF)

**S5 Table. Summary of average laser ablation U-series ages for faunal remains from Leang Bulu Bettue.**
(PDF)

**S6 Table. $D_e$ estimation details for age determination.** Equivalent dose ($D_e$) estimation model, cutoff value used for outlier rejection, number of micro-aliquots accepted during data analysis (after outlier rejection) out of the total measured,

overdispersion values of the $L_n/T_n$ or $D_e$ distribution (depending on age model used) before ($OD_{total}$) and after outlier rejection ($OD_{nMAD}$), expected scatter of a well-bleached $D_e$ distribution ($\sigma_b$), $D_e$ and ages are provided for each sample.
(PDF)

**S7 Table. Sampling and $D_e$ estimation details.** Burial depths and stratigraphic layers, grain sizes, proportion of water per dry sample mass used for age calculation and field water contents in parentheses as well as dosimetry components are provided for each sample. Bulk radionuclide concentrations were determined through thick-source alpha counting coupled with beta-counting. The total dose rate ($\dot{D}_{total}$) is the sum of gamma ($\dot{D}_\gamma$), external beta ($\dot{D}_{\beta\ external}$), internal beta ($\dot{D}_{\beta\ internal}$) and cosmic ($\dot{D}_{cosmic}$) dose rates.
(PDF)

**S8 Table.** Number of Identified Specimens (NISP) and Minimum Number of Individuals (MNI).
(PDF)

**S9 Table.** List of proboscidean fossils from Leang Bulu Bettue.
(PDF)

**S10 Table. Size ranges of lower molars of *Elephas maximus* and other Elephantini species.**
(PDF)

**S11 Table. Measurements of the Leang Bulu Bettue mandible fragments and *Elephas maximus.***
(PDF)

**S1 Fig. Grainsize and ICP-AES results from section -C2.**
(PDF)

**S2 Fig. Grainsize results from section -H2.**
(PDF)

**S3 Fig. Grainsize results from section A1.**
(PDF)

**S4 Fig. Examples of distributions of micro-aliquot (A) SGC $D_e$ or (B) re-normalised $L_n/T_n$ with details of the age models used for dating.** Two age models were used: (A) a central age model (CAM), and (B) a minimum age model (MAM) using $\sigma_b = 0.35$. Outliers were removed prior to either age models using nMAD cutoff values of (A) 1.5 or (B) 2.0. The radial plots are centred on the nMAD CAM and the grey bands are centred on the modelled values (nMAD CAM or nMAD MAM). OD refers to the overdispersion value.
(PDF)

## Acknowledgments

For facilitating this research, we thank former Puslit Arkenas (now BRIN) director I.M. Geria and current director H. Jogaswara, as well as the former director of Makassar's Balai Pelestarian Kebudayaan (BPK), L. Aksa. Fieldworkers included: D. Susanti, R. Salempang, H. Arsyad, Muhtar, Sungkar, R. Ali, L. Lantik, Asri, H. Lahab, O. Amrullah, Idrus, M. Husain, Busran, and A. Agus. For advice and assistance, we thank I. Glover, J. Joordens, W. Roebroeks, C. Little, D. Bulbeck, P. Piper, X. Zhihong, A. Abdul, F. Petchey, U. Pietrzak, L. Kinsley, J. Hellstrom, P. Bajo, S. Wroe, M. Young and R. Klaebe. We acknowledge the Sulawesi universities of Hasanuddin and Haluoleo for their collaboration on fieldwork. We also thank all the people of Leang Leang area of Maros who supported us throughout this decade-long

project. The excavations were authorised by Indonesia's State Ministry of Research and Technology (RISTEK) and its succeeding organisation, the National Research and Innovation Agency (BRIN). Research was conducted in formal collaboration with BRIN (formerly Pusat Penelitian Arkeologi Nasional [ARKENAS]). All necessary permits were obtained from RISTEK-BRIN for the described study (Permit Numbers: 3/TKPIPA/FRP/SM/III/2013, 154/SIP/FRP/E5/Dit.KI/VII/2017, 129/SIP/IV/FR/3/2023, 490/KE.01/SK/06/2024), which complied with all relevant regulations. The archaeological assemblages from LBB are permanently stored at the premises of BRIN's local branch in Makassar (formerly Balai Arkeologi Sulawesi Selatan). Requests to access collections for study, including digital databases and catalogues of archaeological finds, should be directed in the first instance to the head of Pusat Riset Arkeologi Lingkunan, Maritim, dan Budaya Berkelanjutan (BRIN) (pralmbb@brin.go.id) and AB (a.brumm@griffith.edu.au).

## Author contributions

**Conceptualization:** Basran Basran, Budianto Hakim, Iwan Sumantri, Hasliana Hasliana, Linda Siagian, Nur Ihsan Djindar, Marlon N. R. Ririmasse, Irfan Mahmud, Maxime Aubert, Adam Brumm.

**Data curation:** Basran Basran, Budianto Hakim, Iwan Sumantri, Suryatman Suryatman, Adhi Agus Oktaviana, Hasliana Hasliana, Linda Siagian, Nur Ihsan Djindar, Marlon N. R. Ririmasse, Yinika L. Perston, Mark W. Moore, Mariana Sontag-González, Bo Li, Ian Moffat, Brian Jones, Adam Brumm.

**Formal analysis:** Budianto Hakim, Iwan Sumantri, Suryatman Suryatman, Andi Muhammad Saiful, Adhi Agus Oktaviana, Ratno Sardi, Fardi Ali Syahdar, Nur Ihsan Djindar, Yinika L. Perston, Mark W. Moore, Mariana Sontag-González, Bo Li, Gerrit D. van den Bergh, Maxime Aubert, Rainer Grün, David P. McGahan, Michelle C. Langley, Tiina Manne, Ian Moffat, Brian Jones.

**Funding acquisition:** Marlon N. R. Ririmasse, Yinika L. Perston, Mark W. Moore, Adam Brumm.

**Investigation:** Basran Basran, Budianto Hakim, Iwan Sumantri, Suryatman Suryatman, Andi Muhammad Saiful, Adhi Agus Oktaviana, Ratno Sardi, Hasliana Hasliana, Muhammad Ramli, Linda Siagian, Andi Jusdi, Abdullah Abdullah, Fardi Ali Syahdar, Hamrullah Hamrullah, Imran Ilyas, Putra Hudlinas Muhammad, Sofyan Setia Budi, Nur Ihsan Djindar, Shinatria Adhityatama, Rustan Lebe, Marlon N. R. Ririmasse, Irfan Mahmud, Akin Duli, Yinika L. Perston, Mark W. Moore, Mariana Sontag-González, Bo Li, Gerrit D. van den Bergh, Maxime Aubert, Rainer Grün, David P. McGahan, Michelle C. Langley, Emma C. James, Tiina Manne, Ian Moffat, Brian Jones, Adam Brumm.

**Methodology:** Basran Basran, Budianto Hakim, Iwan Sumantri, Suryatman Suryatman, Hasliana Hasliana, Akin Duli, Yinika L. Perston, Mark W. Moore, Adam Brumm.

**Project administration:** Basran Basran, Budianto Hakim, Iwan Sumantri, Andi Muhammad Saiful, Adhi Agus Oktaviana, Hasliana Hasliana, Muhammad Ramli, Linda Siagian, Nur Ihsan Djindar, Shinatria Adhityatama, Rustan Lebe, Marlon N. R. Ririmasse, Irfan Mahmud, Akin Duli, Yinika L. Perston, Mark W. Moore, Maxime Aubert, David P. McGahan, Adam Brumm.

**Resources:** Budianto Hakim, Hasliana Hasliana, Sofyan Setia Budi, Nur Ihsan Djindar, Rustan Lebe, Marlon N. R. Ririmasse, Mark W. Moore, Maxime Aubert, Ian Moffat, Adam Brumm.

**Software:** Ratno Sardi, Nur Ihsan Djindar.

**Supervision:** Budianto Hakim, Iwan Sumantri, Suryatman Suryatman, Hasliana Hasliana, Linda Siagian, Nur Ihsan Djindar, Irfan Mahmud, Akin Duli, Mark W. Moore, Gerrit D. van den Bergh, Michelle C. Langley, Adam Brumm.

**Validation:** Hasliana Hasliana, Marlon N. R. Ririmasse, Irfan Mahmud.

**Visualization:** Basran Basran, Hasliana Hasliana, Nur Ihsan Djindar, Adam Brumm.

**Writing – original draft:** Basran Basran, Budianto Hakim, Iwan Sumantri, Suryatman Suryatman, Andi Muhammad Saiful, Adhi Agus Oktaviana, Ratno Sardi, Hasliana Hasliana, Muhammad Ramli, Andi Jusdi, Abdullah Abdullah, Fardi Ali Syahdar, Hamrullah Hamrullah, Imran Ilyas, Putra Hudlinas Muhammad, Nur Ihsan Djindar, Shinatria Adhityatama, Rustan Lebe, Marlon N. R. Ririmasse, Akin Duli, Yinika L. Perston, Mark W. Moore, Mariana Sontag-González, Bo Li, Gerrit D. van den Bergh, Maxime Aubert, Rainer Grün, Emma C. James, Ian Moffat, Brian Jones, Adam Brumm.

**Writing – review & editing:** Basran Basran, Yinika L. Perston, Mark W. Moore, Mariana Sontag-González, Adam Brumm.

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
