## [Decision Letter · Decision Letter 0]

17 Nov 2025

A near-continuous archaeological record of Pleistocene human occupation at Leang Bulu Bettue, Sulawesi, Indonesia

PONE-D-25-58502

Dear Dr. Barsan,

We’re pleased to inform you that your manuscript has been judged scientifically suitable for publication and will be formally accepted for publication once it meets all outstanding technical requirements.

Your paper is of high importance and I am sorry for the major delay in its publication. Ifound tge revisions made satisfactory and the paper fitts plos one very well.

Kind regards,

Ran Barkai

Academic Editor

PLOS ONE

Journal Requirements:

1. Please include a complete copy of PLOS’ questionnaire on inclusivity in global research in your revised manuscript. Our policy for research in this area aims to improve transparency in the reporting of research performed outside of researchers’ own country or community. The policy applies to researchers who have travelled to a different country to conduct research, research with Indigenous populations or their lands, and research on cultural artefacts. The questionnaire can also be requested at the journal’s discretion for any other submissions, even if these conditions are not met.  Please find more information on the policy and a link to download a blank copy of the questionnaire here: https://journals.plos.org/plosone/s/best-practices-in-research-reporting. Please upload a completed version of your questionnaire as Supporting Information when you resubmit your manuscript.

2. In your manuscript, please provide additional information regarding the specimens used in your study. Ensure that you have reported human remain specimen numbers and complete repository information, including museum name and geographic location.

'All necessary permits were obtained for the described study, which complied with all relevant regulations.

For more information on PLOS One's requirements for paleontology and archeology research, see https://journals.plos.org/plosone/s/submission-guidelines#loc-paleontology-and-archaeology-research .

Australian Research Council grants DE130101560 and FT160100119 awarded to A.B. and DE160100703 awarded to I.M., with additional support provided by the Wenner Gren Foundation (Post-Ph.D. grant) and Griffith University.

Please respond by return e-mail so that we can amend your financial disclosure and competing interests on your behalf.

Additional Editor Comments (optional):

This is a very important contribution to the field of Archaeology and human manifestations. I strongly support it's publication in plus one. The revisions done are satisfactory, and the additional review is provided as food for thought

Reviewers' comments:

Reviewer's Responses to Questions

**Comments to the Author**

1. Is the manuscript technically sound, and do the data support the conclusions?

Reviewer #1: Yes

2. Has the statistical analysis been performed appropriately and rigorously?

Reviewer #1: N/A

3. Have the authors made all data underlying the findings in their manuscript fully available?

Reviewer #1: Yes

•Svenning, J.-C. et al. (2024): Provides a comprehensive overview of megafauna extinction patterns, causes, and ecological consequences, offering essential background for understanding global ecosystem shifts.

•Barkai, R. et al. (2024): Discusses the absence of cave art in the Upper Paleolithic Southern Levant and explores links between megafaunal decline and changes in symbolic expression.

•Ben-Dor, M. & Barkai, R. (2025): Proposes a hypothesis connecting megafaunal extinction with bioenergetic-driven cultural and behavioral adaptations, including the rise of symbolic artifacts.

The manuscript is otherwise well written and suitable for publication.

•Svenning, J.-C. et al. The late-Quaternary megafauna extinctions: patterns, causes, ecological consequences, and implications for ecosystem management in the Anthropocene. Cambridge Prisms: Extinction, 1–68 (2024).

•Barkai, R., Dagoni, I., Ben-Dor, M. & Kedar, Y. Why is Cave Art Absent from the Upper Paleolithic Southern Levant? Mitkufat Haeven - Journal of the Israel Prehistoric Society 54, 177–202 (2024).

**Do you want your identity to be public for this peer review?** For information about this choice, including consent withdrawal, please see our Privacy Policy

---

## [Editor Report · Acceptance letter]

PONE-D-25-58502

PLOS One

Dear Dr. Basran,

I'm pleased to inform you that your manuscript has been deemed suitable for publication in PLOS One. Congratulations! Your manuscript is now being handed over to our production team.

Kind regards,

on behalf of

Professor Ran Barkai

Academic Editor

PLOS One